# Financial development, technological innovation and urban-rural income gap: Time series evidence from China

Li-Min Wang[1,2], Xiang-Li Wu[1,2]*, Nan-Chen Chu[1,2]*

**1** College of Geographical Sciences, Harbin Normal University, Harbin, China, **2** Key Laboratory of Remote Sensing Monitoring of Geographic Environment of Heilongjiang Province, Harbin Normal University, Harbin, China

* jndxwxl@163.com (XLW); chunanchen_1992@163.com (NCC)

**Data Availability Statement:** All relevant data are within the paper and its Supporting information files.

**Funding:** This work was supported by the Key Plan of National Social Science Foundation of China

## Abstract

The main purpose of the paper is to investigate the relationship between technological innovation and income inequality for China based on the financial Kuznets curve (FKC) hypothesis. The study uses time-series data from 1985 to 2019. We employ the Johansen cointegration, ARDL model and VECM Granger causality techniques to analyze the links between the variables. We also use the DOLS, FMOLS and CCR mechanisms to estimate the long-run parameters. The paper finds that the FKC is valid for China's economy in the long run. Technological innovation positively affects the urban-rural income gap, while there is an inverted-U shaped between financial development and the urban-rural income gap. The relationship between financial development and the urban-rural income gap is bi-directional causality. Technological innovation and the urban-rural income gap cause each other. Empirical results suggest a twofold policy meaning: i) to further the financial system and ii) to eliminate the adverse impacts of technological innovations on income distribution.

## Introduction

Income distribution has always been a popular issue in entire society. A reasonable income distribution system and income gap are not only the common aspiration of the people but also an important embodiment of social justice. China's urban-rural dual structure problem results from the country's policy choice in the process of reform and opening up. The country's city priority development strategy and the opening-up pilot program of coastal cities and regions have attracted an influx of talents and funds from inland areas, which has led to further disparities in income distribution between urban and rural areas and regions. Especially in recent years, with the widening gap between urban and rural residents' income growth rate, the income gap between urban and rural areas in China continues to increase, which has become an obstacle to the long-term healthy and stable development of China's economy. The data from China Statistical Bureau show that the income ratio of urban and rural residents in China has decreased from 3.30: 1 in 2007 to 2.64: 1 in 2019, and the overall trend is declining. However, compared with the world level, the income gap between urban and rural areas in

under the Grant 16BJY039. The funders had role in study design, data collection and analysis, decision to publish, or preparation of the manuscript.

**Competing interests:** The authors declare that they have no known competing financial interests or personal relationships that could have appeared to influence the work reported in this paper.

China is hovering at a high level and shows a fluctuating trend. During 2002–2019, it expanded at a high level before 2009, and then fell within a narrow range, showing an inverted U-shaped trend. The Fifth Plenary Session of the 19th Central Committee of the Communist Party of China proposed the long-term goals for 2035, including 'more obvious and substantial progress in common prosperity for all people' and 'per capita GDP reaches the level of moderately developed countries, and the middle-income group has significantly expanded'. At present, the government has introduced a variety of agricultural financial policies to promote rural economic development, and gradually relaxed the household registration management system to strengthen labor mobility. However, under the current urban-rural dual structure system, the income distribution of residents in China is still very serious, and the income gap between urban and rural residents is expanding, which will inevitably affect social harmony and stability. Therefore, in realizing the path of establishing and improving the institutional mechanism of urban-rural integration development and balancing urban-rural integration development put forward in the 19th National Congress of the Communist Party of China, solving the urban-rural income gap has become the key point the sustainable economic development at this stage and in the future for a long time.

Following Kuznets [1], Greenwood and Jovanovich [2] first reveal an inverted U-shaped relationship between financial development and income inequality. Namely, with the development of finance, the income gap shows an inverted U-shaped feature of early expansion and then reduction, which is the financial Kuznets curve (FKC) hypothesis. In the early 1990s, the income gap between countries in the world was highlighted, and the mechanism of financial development on income distribution attracted wide attention in academia. At present, the research in this area has become a research frontier of financial development theory. Looking at the actual situation of our country, since the reform and opening up, the financial market has developed rapidly, the reform of the financial industry has achieved positive development, and the innovation of the industry has gradually deepened. In 2018, the scale of social financing reached 200.75 trillion yuan in the entire year, and the scale of the interbank bond market leaped to the second in the world. However, with the rapid development of financial markets, accompanied by the widening income gap between urban and rural residents, the problem of unbalanced and inadequate development still exists. Furthermore, technological innovation is considered to be the major factor affecting macroeconomic variables. Hou Zhenmei, Tian Mao, et al. [3] pointed out that the process of technological innovation is an economic growth process that takes technology as the mainline and causes social change and technological innovation is a key driving force for economic growth, and its development process will inevitably affect the change of income distribution pattern of different groups. Pan et al. [4], Wang and Wang [5] dwell on the influence of technological innovation on energy efficiency. Technological innovation can influence environmental degradation. Recently, the effect of innovation on the income gap is an important field of research [6].

The China's economy is an essential case. In 2016, China became the first middle-income economy to enter the top 25 of the global innovation index. China's scientific and technological innovation capability was enhanced and the main scientific and technological innovation indicators were steadily improved in 2018. The state intellectual property office of the data shows the proportion of research and experimental development expenditure of the entire society in GDP in 2018 was 2.15%. The total number of R&D personnel reached 4.18 million person-years, ranking first in the world; The number of international scientific papers and citations ranked second in the world; The number of patent applications and authorization of invention ranks first in the world; The contribution rate of scientific and technological progress is expected to exceed 58.5%, and the national comprehensive innovation ability ranks 17th in the world. The level of scientific and technological innovation in China has been continuously

improved, and fruitful achievements have been made in many fields such as innovation input, innovation output, and innovation efficiency. These developments show technological innovations advance rapidly in China. In addition, income distribution has been a vital matter in China for a long time. The report of the 19th National Congress of the Communist Party of China pointed out that my country's 'urban and rural regional development and income distribution gap is still large', which must be solved. In this context, to incorporate technological innovation into the important factors that affect the income gap between urban and rural areas, and to explore ways and channels to ease and eliminate the excessive income gap in residential areas in China, which is of great significance to reduce income inequality and realize social equity at this stage.

Considering all the above explanations, the present study creates the following questions: First, is the FKC hypothesis valid for China? Second, how technological innovation influences income gap in China's economy? Third, is there a causal linkage between technological innovation and urban-rural income gap? In addition, there are no studies on the causal linkage between them. In this regard, this is the first study on China's economy. To answer these questions above, we attempt to make an empirical study on the link between technological innovation and income gap in China from 1985 to 2019 under the context of FKC hypothesis. We intensify on the long-term and causal relations between the variables. In the study of financial development and the income gap between urban and rural residents, this paper adds the index of technological innovation. By citing these three variables, hope to be more comprehensive on the causes of China 's urban-rural income gap and how to solve this problem are discussed, and thus draw more accurate results and set forward more constructive suggestions. The widening income gap between urban and rural areas in China has attracted wide attention from the government and academia, but from the domestic existing literature, it focuses on the role of law decision-making factors. There is no analysis on the impact of technological innovation on the income gap under the background of FKC hypothesis. In addition, the utility research on the long- and short-term causality test between technological innovation and income gap is indeed more inadequate. Thus, the study intends to focus on verifying the validity of the FKC hypothesis and the dynamic relationship between technological innovation and the urban-rural income gap in the FKC hypothesis. It provides the theoretical basis and policy suggestions for further deepening rural financial reform and technological innovation and giving full play to the role of financial services and technological innovation in the allocation of rural resources. This newly explains the causes of the urban-rural income gap, which is not only related to the overall promotion of the coordinated development of urban and rural areas but also the determination of the focus and specific direction of future financial and technological innovation development in various regions. Therefore, it is an important subject worthy of in-depth study and testing.

## An overview of the literature

The urbanization process and financial development of developed countries began earlier. Foreign scholars have carried out relatively full research on financial development and income gap, but no consensus has been reached and there is a big controversy. FKC Hypothesis was first developed by Kuznets [1], an American economist and statistician, through an analysis of data related to economic growth and income gap in several countries, there is an inverted U-shaped relationship between financial development and urban-rural income gap. This is because, in the early stage of financial development, only a few high-income people can pay the cost of financial services and get high returns financial services, so the income gap expands; When the financial development is to a certain extent, most people can cross the wealth

threshold of financial services, share the benefits of financial development, the income gap narrowed slowly. Then Greenwood and Jovanovic [2] found developed a GJ model to explore the communication between financial development and economic growth and income distribution and found that they also conformed to the 'inverted U' trend. Townsend and Ueda improved the model on this basis and reached similar conclusions [7]. However, Clarke [8] adjusted the research data to the panel data of several countries and found that from an international perspective, financial development and income gap did not show an 'inverted U' feature, but were negatively correlated. Further deepening of financial development would have a positive effect on narrowing the urban-rural income gap. Pradhan [9] used time-series data from India 45 to show the relationship between economic growth, financial development, and the income gap between urban and rural residents, and also found that financial development is conducive to narrowing the income gap. However, some scholars have reached different conclusions in their studies. Law et al. [10] used the threshold regression method to conduct an empirical analysis of financial development and income gap. The results show that there is a threshold effect between them. Only after reaching a certain threshold, financial development will narrow the income gap, and the implementation does not exist before. Sehrawat and Giri use ARDL model analysis results show that both in the long and short term. India's financial development level has increased the income gap [11].

Since the 1990s, with the swift growth of financial markets and the advance of urbanization, the class bias of communal income has come better and further serious. Domestic scholars analyzed and discuss the issue of the urban-rural income gap. Many scholars have different perspectives on the impact of financial development on the urban-rural income gap. For example, Qiao Haishu and Chen Li [12], Hu Zongyi and Liu Yiwen [13], and Yang Nan and Ma Chuoxin [14] upheld that the impact mechanism of China's financial development on the urban-rural income gap is also in line with the 'inverted U' component. Based on the theory of economic growth, Ye Zhiqiang et al. [15] analyzed and discussed the impact of financial development on narrowing the income gap, and thought that the problem of poverty in rural China cannot be solved effectively by developing finance alone, and unreasonable financial expansion will lead to the widening of the income gap between urban and rural areas. Jia Fei found that the low financial transformation efficiency in rural areas will lead to the imbalance of urban and rural financial development and further aggravates the widening income gap [16]. From the standpoint of the regional economy, Sun Yongqiang et al. [17] pointed out that for a long time, China's financial development and financial development showed a significant positive correlation, especially in the eastern region. Yang Youcai used the threshold model to verify the threshold effect of financial development and found that the financial threshold of China's regions showed an increasing trend from west to east. However, due to the differences between the corresponding thresholds of the level of financial development in the eastern and western regions, there are also differences in the changes in the income gap caused by financial growth in the eastern and western regions [18].

Existing studies have shown that technological innovation is a key determinant of the urban-rural income gap. Technological innovation promotes economic development and pushes the development process of industrialization and urbanization. What is the impact of technological innovation on the income gap? Foreign scholars have conducted in-depth research on the impact of technological innovation on income disparity and the results of the study are also different. The major research results focus on the following aspects:

Firstly, the skill bias of technological progress makes the pay growth of high-skilled and low-skilled workers polarization, which leads to the widening income gap, the effect of skill-biased technological progress. Leamer believes that technological progress in technology-intensive sectors contributes to higher wages for skilled workers, in labour-demanding parts

contributes to higher wages for unskilled workers, and that technological progress in technology-intensive sectors has a more pronounced impact on income gaps [19].

Second is the spillover effect of skills. Aghion and others found new technologies can improve labor productivity, and workers can learn to increase their knowledge [20]. The result is that the wages of workers in the same skill level group are rare, and the relative supply and technology spillover efficiency of skilled workers among workers various skill levels cause the wage gap between workers with different skill levels. Cirillo.V analysis shows that technological innovation is conducive to narrowing the overall income gap, but it will aggravate the income gap among high-income groups [21].

The third is the Organizational Effect on Labor Market. Nathan M points out that technological innovation affects the direction and intensity of the income gap, which is also related to aspects such as the size of the labor force, the structure of labor skills and the degree of economic development [22]. Akcigit et al. [23] remarked that innovation exacerbates the gap between high-income meets and that the income of most successful innovators has risen dramatically.

Domestic scholars have also studied this issue from multiple perspectives, aiming at the relationship between technological innovation and income gap. Some studies believe that technological innovation tends to widen the income gap. Chen Yong and Bai Zhe found that skill-biased technological progress is the most important factor in the widening regional wage gap in China [24]. Zeng Peng et al. [25] found that in terms of China's urban agglomerations, technological progress will expand the urban-rural income gap, and technological progress will promote the improvement of urbanization. However, some scholars believe that technological innovation can alleviate the income gap. Dong Zhiqing et al. [26] and others believe that neutral technological advances can raise the supply of skilled labour and reduce the wage gap between skilled and non-skilled labour. Ma Lei discussed the impact of total factor productivity and human capital structure on urban-rural income gap from the perspective of innovation-driven development [27]. The study found that the optimization of human capital structure in the central and western regions has a powerful effect on reducing the urban-rural income gap. The growth of total factor productivity and the improvement of technological progress can narrow the gap in the central and western regions, but it shows an expansion effect in the eastern field.

From the above summary, we can see that the existing literature has conducted in-depth research on the problem of urban-rural income gap, and applied research has also made abundant achievements. However, there are still some shortcomings which need to be further studied: Firstly, the empirical results related to FKC hypothesis are very complex and inconsistent. The main reason for this statement is that periods, analysis techniques and country groups considered are different. Secondly, most of the studies focus on the validity of FKC hypothesis, but they do not integrate technological innovation into the income gap specifications. Lastly, even if several studies only investigate the technological innovation-income gap link, they do not dwell on the FKC hypothesis. The research on the above three points can be regarded as the contribution of this paper. Therefore, the present study aims at examining the relationship between technological innovation and urban-rural income gap in the presence of FKC hypothesis for China's economy, and put forward policy recommendations on this basis.

## Materials and method

### The selection of variables

**Technological innovation.** The technological innovation process of a country is embodied in the economic growth process that takes technology as the main line and causes social

changes. The process of technological innovation and development will inevitably affect the changes of income distribution pattern of different groups. In measuring the level of technological innovation, there are usually two interrelated patent indicators, namely the amount of patent authorization and the amount of patent acceptance, which are widely used in the literature. Compared with the patent acceptance, the index of patent authorization can more calculate the innovation level. Therefore, this paper will still use this index to measure the level of technological innovation in China.

**Financial development.** Goldsmith proposed that the financial correlation ratio refers to the ratio of the value of all financial assets of a country to the total amount of economic activities in that country at a certain date. The change in financial related ratio reflects the changing relationship between the financial sector and the economic base in terms of scale, which is used to measure the scale of a country's or region's financial development and the degree of financial deepening. Based on the fact that China's financial market is not perfect and the scale of the banking industry accounts for a high proportion of the financial industry, So this study selects the Gohren index (FIR) which is measured by the ratio of the end-of-year loans of financial institutes to GDP. Since the inverted U-shaped contact between the level of financial development and the income gap has regularly been a hot subject in academic research. Scholars all use this variable as an important factor in analyzing income disparity. Therefore, this paper introduces the financial development scale (FIR) and its quadratic call to demonstrate whether there is an inverted U-shaped link between the two.

**Urban-rural income gap.** Indicators to measure the income gap between urban and rural residents are Gini coefficient, Theil coefficient, Wolfson polarization index, etc. Based on the pertinent indexes selected in this article need to reflect the efficiency of the input and output level of urban and rural residents' productivity, so this paper selects the urban residents' per capita disposable income UI and the ratio of rural residents' per capita net income RI to measure the income gap between urban and rural residents.

## Data source and processing

The data utilized in this empirical analysis is the time series data of the whole country from 1985 to 2019, the per capita disposable income of urban residents, the per capita net income of rural residents, GDP, the end-of-year loan amount of financial institutions and the amount of patent authorization. The data are collected from the China Statistical Yearbook of Science and Technology, National Economic and Social Development Statistics Bulletin of the People's Republic of China, China Statistical Yearbook, Compilation of Statistics for 60 Years in New China and the official network of the individual's Bank of China.

In the process of data processing, in order to eliminate the endogenous and heteroscedasticity of variables, natural logarithm processes the variable sequence. Eviews10.0 and Stata16.0 complete the subsequent empirical research and analysis in this paper. The statistical description of the data (see Table 1) and the time trend change (see Fig 1) are as follows.

**Table 1. Descriptive statistics (1985–2019).**

| | Mean | Median | Maximum | Minimum | Std. Dev | Skewness | Kurtosis | Jarque-Bera | Observations |
|---|---|---|---|---|---|---|---|---|---|
| lnGI | 1.003809 | 1.004728 | 1.203820 | 0.619987 | 0.149692 | -0.548122 | 2.691849 | 1.891034 | 35 |
| lnTI | 11.80701 | 11.79358 | 14.76779 | 4.927254 | 2.166763 | -0.859063 | 4.120165 | 6.134806 | 35 |
| lnFIR | 4.639338 | 4.620645 | 5.036837 | 4.221308 | 0.204137 | 0.242914 | 2.430347 | 0.817446 | 35 |
| lnFIR$^2$ | 21.56394 | 21.35036 | 25.36972 | 17.81944 | 1.904705 | 0.330961 | 2.440912 | 1.094800 | 35 |

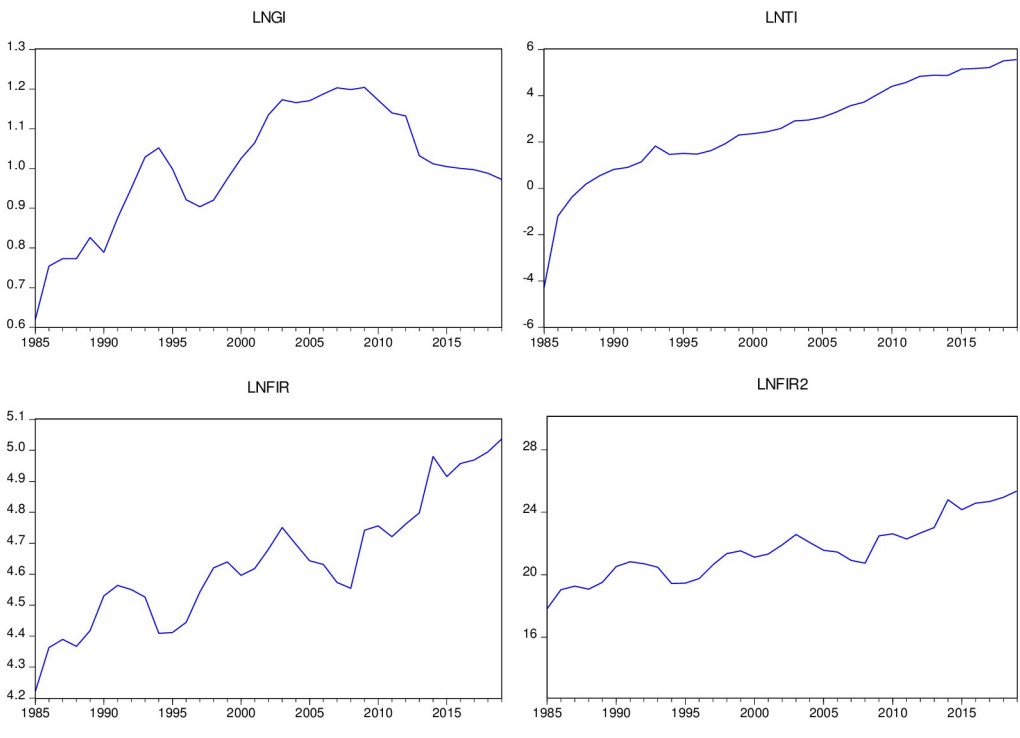

**Fig 1. Trends in the series (1985–2019).**

Fig 1 shows the time series of the natural logs of the urban-rural income gap from 1985 to 2019, technological innovation, financial development and its squares (lnGI, lnTI, lnFIR, lnFIR$^2$, respectively). China's urban-rural income gap (lnGI) indicates an upward trend until the mid-1990s, followed by a brief decline, and then continued to maintain growth. It peaked in 2009 with a maximum value of 1.204, and has since been on a downward trend until it fell to 0.972 in 2019. This shows that the variable resembles an inverted-U type of shape. After 2013, the income gap between urban and rural residents has stabilized, which may be due to the gradual improvement of the urban and rural social security system and the income of rural residents has increased significantly. So the income gap between urban residents and urban citizens has gradually narrowed. In addition, my country's technological innovation continues to maintain a growth trend; Although financial development variables show volatilities in the period, the variables in general tend to increase. The proxies for financial development (lnFIR) have average values of 4.369 in that period respectively while maximum values were obtained in 2019 with 5.037. While financial development indexes show a positive trend, urban-rural income gap indicators having an inverse-U-shaped tend, which nurtures the motivation to study the linkages between financial development and urban-rural income gap under the background of the financial Kuznets curve.

## Model construction and research methods

**Basic model setting.** This paper focuses on the effect of technological innovation on the urban-rural income gap in China's economy for the period 1985–2019. The study's results, more focused on the role of financial development factors, and the important technological innovation element in the cause of the urban-rural income gap, have not been fully analyzed. Therefore, we integrate technological innovation into the income gap equations by taking into

account the FKC hypothesis. Thus, the augmented income inequality functions can be modelled as follows:

$$\ln GI = C + \beta_1 \ln TI + \beta_2 \ln FIR + \beta_3 \ln FIR^2 + \delta \tag{1}$$

lnGI indicates the income gap between urban and rural residents; lnTI as a level of technological innovation. lnFIR and $\ln FIR^2$ indicators of financial development level; C denote the constant; $\beta_1 \sim \beta_4$ indicates the parameter to be estimated, $\delta$ is the error term.

**Method.** The empirical strategy of this study mainly consists of four different stages. We first apply the ADF and PP unit root tests presented by Dickey and Fuller, Phillips and Perron and Elliott et al. for stationary analysis [28–31]. Second, the cointegration methods proposed by Pesaran et al. are used to determine the cointegration between the series [31]. Third, the long-run coefficients are estimated through the DOLS, FMOLS and CCR estimation methods developed by Stock and Watson, Phillips and Hansen and Park, respectively [32–34]. In the last stage, the causal relations between the series are analyzed by the VECM Granger causality method presented by Engle and Granger [35].

**Autoregressive distributed lag model.** In this paper, the ARDL method for cointegration test has the following two obvious advantages. First, the approach does not require the variable data to be a single integer sequence of the same order. After determining the optimal lag order, ARDL can analyze the long-term relationship of variables regardless of whether the variables are I(0), I(1) or mixed sequences. Another advantage is that we can derive the dynamic error correction model to analyze the long-term and short-term dynamic effects of time series [36]. ARDL is modeled as follows:

$$
\begin{aligned}
\triangle \ln GI = {} & C + \sum_{k=1}^{n} \lambda_{1k} \triangle \ln GI_{t-k} + \sum_{k=0}^{n} \lambda_{2k} \triangle \ln FIR_{t-k} \\
& + \sum_{k=0}^{n} \lambda_{3k} \triangle \ln FIR^2_{t-k} + \sum_{k=0}^{n} \lambda_{4k} \triangle \ln TI_{t-k} + \gamma_1 \ln GI_{t-1} \\
& + \gamma_2 \ln FIR_{t-1} + \gamma_3 \ln FIR^2_{t-1} + \gamma_4 \ln TI_{t-1} + u_r
\end{aligned}
\tag{2}
$$

Where, $\triangle$, k, C and $u_r$ indicate the first-order difference, the lag order of difference term, the constant term and the white noise error term, respectively. In the ARDL bounds test, the null and alternative hypotheses are as follows:

$$H_0 = \gamma_1 = \gamma_2 = \gamma_3 = \gamma_4 = 0 \tag{3}$$

$$H_1 \neq \gamma_1 \neq \gamma_2 \neq \gamma_3 \neq \gamma_4 \neq 0 \tag{4}$$

In this procedure, we first select the optimal lag length through the AIC or SBC. Pesaran et al. proposed the F-statistic to examine the cointegration between the variables [31]. Second, we compare the F-statistic with the critical bounds tabulated by Pesaran et al. [31]. When the F-statistic is higher than the upper critical bounds (UCB), there is a long-run relation between the series. When the F-statistic is smaller than the lower critical bounds (LCB), we conclude that there is no cointegration between the series. If the F-statistic is between the UCB and LCB, we cannot make an interpretation of the finding. The ARDL bounds test is preferable to the other classical cointegration tests.

**Co-integration equation estimation: FMOLS, DOLS, CCR.** After testing the cointegration relationship between variables, the long-run coefficients between variables are estimated by DOLS, FMOLS and CCR estimation technique, respectively. The DOLS technique is an asymptotically effective estimator that eliminates autocorrelation, simultaneity and endogeneity

problems in the cointegration equation. The FMOLS has the advantage of correcting for auto-regression and endogeneity problem. The CCR is a very simple to apply and based on a transformation of the variables in the co-integrating regression [37].

*VECM Granger causality test*. In the last stage of the empirical analysis, the VECM Granger causality test is employed for causality. Engle and Granger add the error correction term (ECT) in the classical VAR model as an additional variable [35]. Therefore, it has the advantage of being able to analyze the long-term and short-term causality between variables. The Granger causality test based on the error correction term (ECT) is carried out through the vector error correction model. In our empirical specifications the VECMs can be written:

$$
\begin{bmatrix} \triangle \ln GI_t \\ \triangle \ln FIR_t \\ \triangle \ln TI_t \end{bmatrix} = \begin{bmatrix} \alpha_1 \\ \alpha_2 \\ \alpha_3 \end{bmatrix} + \begin{bmatrix} A_{11,1} A_{12,1} A_{13,1} \\ A_{21,1} A_{22,1} A_{23,1} \\ A_{31,1} A_{32,1} A_{33,1} \end{bmatrix} \times \begin{bmatrix} \triangle \ln GI_{t-1} \\ \triangle \ln FIR_{t-1} \\ \triangle \ln TI_{t-1} \end{bmatrix} + \ldots
$$
$$
+ \begin{bmatrix} A_{11,n} A_{12,n} A_{13,n} \\ A_{21,n} A_{22,n} A_{23,n} \\ A_{31,n} A_{32,n} A_{33,n} \end{bmatrix} \times \begin{bmatrix} \triangle \ln GI_{t-1} \\ \triangle \ln FIR_{t-1} \\ \triangle \ln TI_{t-1} \end{bmatrix} + \begin{bmatrix} \theta_1 \\ \theta_2 \\ \theta_3 \end{bmatrix} \times (ECT_{t-1}) + \begin{bmatrix} \varepsilon_1 \\ \varepsilon_2 \\ \varepsilon_3 \end{bmatrix} \tag{5}
$$

Where, $\triangle$, $\varepsilon_t$, $\alpha$ and i indicate the difference operator, the white noise error term, the constant term and the optimal number of lag, respectively. The matrix of A serves as the long-run coefficients of the series. $ECT_{t-1}$ is the lagged residual term derived from the long-run relationship. The long-run causality causes the influence of the t-statistic related to the coefficient of the error correction term ($ECT_{t-1}$) while the short-run causality requires the significance of the F-statistic in the first differences of the variables.

## Empirical results and discussion

### Unit root test

To avoid the phenomenon of pseudo-regression, the variables of the time series must be stable before constructing the dynamic econometric model. This study uses ADF and PP as two test methods to test the unit root stationery of the time series of the variables one by one. From the test results (see Table 2), at the significance level of 1%, the t statistical values of the ADF and PP tests of lnTI are −16.04485 and −13.7094, which are less than the critical values. Therefore, the null hypothesis is rejected. There is no unit root in lnTI, which is the zero-order unitary sequence I(0), the original sequence is stable. However, the first-order difference series of lnGI, lnFIR and lnFIR$^2$ are stable at a 1% significant level, belonging to the first-order mono integral sequence I(1). In order to analyze the cointegration relationship of the same order difference sequence, we carried the cointegration test out below.

After detecting the stationery of the time series, determining the optimal lag period of the VAR model is very important for the subsequent cointegration test and causality analysis. The selection of the optimal lag period in this paper is determined according to the minimum information criterion of AIC and SC (see Table 3). When the lag period is 1, the values of AIC and SC are the smallest. Therefore, the lag order of the VAR model is 1. It showed the specific results of Johansen cointegration test and ARDL boundary test are shown in Tables 4 and 5.

**Table 2. Unit root test results.**

| Variables | Form | ADF t-statistic | PP Adj.t-statistic | Result |
|---|---|---|---|---|
| lnGI | (C,T,1) | -1.661718 | -2.77091 | - |
| lnTI | (C,T,1) | -16.04485*** | -13.7094*** | I(0) |
| lnFIR | (C,T,1) | -2.745414 | -2.88569 | - |
| lnFIR$^2$ | (C,T,1) | -2.591233 | -2.72779 | - |
| ΔlnGI | (C,T,1) | -4.474620*** | -4.50768*** | I(1) |
| ΔlnFIR | (C,T,1) | -5.519429*** | -8.02233*** | I(1) |
| ΔlnFIR$^2$ | (C,T,1) | -5.56658*** | -8.01153*** | I(1) |

Note: Δ represents the first-order difference of variables; (C,T,K) represents the intercept term, trend term and lag order of ADF respectively;

***represents rejection of the null hypothesis at the 1% significance level.

**Table 3. Optimal lag length selection.**

| Lag | LogL | LR | FPE | AIC | SC | HQ |
|---|---|---|---|---|---|---|
| 0 | 77.5882 | NA | 1.0e-07 | -4.74762 | -4.5626 | -4.6873 |
| 1 | 209.7920 | 264.4100 | 5.7e-11* | -12.2447* | -11.3195* | -11.9431* |
| 2 | 223.4430 | 27.3020 | 7.0e-11 | -12.0931 | -10.4278 | -11.5503 |
| 3 | 232.0850 | 17.2840 | 1.3e-10 | -11.6184 | -9.21301 | -10.8343 |
| 4 | 257.1810 | 50.1910* | 1.0e-10 | -12.2052 | -9.0597 | -11.1799 |

Note:

*indicates the optimal lag length.

## Cointegration test

Since the financial development level (lnFIR) and urban-rural income gap (lnGI) of the original sequence is not stable. To avoid the presence of pseudo regression, we need to co-integration test of these two variables. This paper first uses the EG two-step method Johansen cointegration test (see Table 4). The results show that at the 1% confidence level, the t statistic is -2.636901, which is smaller than -2.636901. The null hypothesis is rejected, and there is a cointegration relationship at the 1% significance level, showing the level of financial development (lnFIR) and urban-rural income cointegration relationship between gaps (lnGI). In addition, the original sequence of technological innovation level (lnTI) is stable, so we can be considered that there is no pseudo-regression among the three variables, and there is a long-term stable equilibrium relationship.

**Table 4. Johansen Cointegration test results.**

| Null Hypothesis: E has a unit root | | | |
|---|---|---|---|
| **Exogenous: None** | | | |
| **Lag Length: 0 (Automatic—based on SIC, maxlag = 1)** | | | |
| | | t-Statistic | Prob.* |
| Augmented Dickey-Fuller test statistic | | -3.110671 | 0.0043 |
| Test critical values: | 1% level | -2.636901 | |
| | 5% level | -1.951332 | |
| | 10% level | -1.610747 | |

**Table 5. Boundary test.**

| Panel A: ARDL results | | | | | |
|---|---|---|---|---|---|
| Estimated models constant(case3) | F-statistic | LM test | ARCH test | Jarque-Bera test | Durbin Watson |
| $\ln GI_t = f(\ln TI_t, \ln FIR_t, \ln FIR^2_t)$ constant and trend(case5) | 4.7120*** | 1.5560[0.2211] | 0.0936[0.7617] | 1.3285[0.5147] | 1.9814 |
| $\ln GI_t = f(\ln TI_t, \ln FIR_t, \ln FIR^2_t)$ | 6.0614** | 1.9051[0.1948] | 0.8665[0.3599] | 1.3686[0.50444] | 1.9689 |
| Panel B: Critical values | | | | | |
| | Constant | | | Constant and trend | | |
| | 1% | 5% | 10% | 1% | 5% | 10% |
| Upper bounds, I(0) | 3.65 | 2.79 | 2.37 | 5.17 | 4.01 | 3.47 |
| Lower bounds, I(1) | 4.66 | 3.67 | 3.20 | 6.36 | 5.07 | 4.45 |

Note: The p-values are given in [].

*** and ** denote significance at 1% and 5% levels, respectively.

Table 5 lists the specific results of the ADRL boundary cointegration test in the two cases of Constant, Constant and Trend respectively. The results show that the F statistical values in the constant, constant and trend items are 4.7120 and 6.0614, respectively, which are greater than the statistics of the upper boundary values at the significance levels of 1%, 5% and 10%. Accordingly, the null hypothesis of no cointegration is rejected at different levels of significance suggesting that there exists the cointegration between the variables. This hints that there is the long-run relationship between the variables. Table 5 also presents the diagnostic tests for the ARDL models: LM test, ARCH test and Jarque-Bera test results suggest that the auto-correlation and heteroscedasticity problems are absent and the error terms are normally distributed. The above analysis proves that the ADRL boundary cointegration test and the Johansen cointegration test have the same results. Therefore, there is a long-term stable equilibrium relationship between technological innovation (lnTI), financial development (lnFIR), financial development (lnFIR$^2$) and urban-rural income gap (lnGI).

## Estimation results

After checking the co-integration between the series, the long-term parameters can be estimated by the FMOLS, DOLS and CCR estimators. The estimation results are shown in Table 6. The coefficients of lnFIR and lnFIR$^2$ are found positive and negative at 1% level of significance, respectively. The findings are consistent with the validity of the FKC hypothesis proposed by Greenwood and Jovanovich [2]. Thus, this reveals that there exists an inverted-U shaped relationship between financial development and the urban-rural income gap in China's economy. Namely, in the initial stage of financial development, the income gap between urban and rural areas continues to expand. When developed to a certain extent, because of the diffusion and infiltration of financial resources from central cities to surrounding rural areas, and improve the financial imbalance. The scale of rural financial resources is expanded and the utilization efficiency is raised. Capital accumulation and technological progress in rural areas promote sustained income growth of rural residents, then narrow the income gap between urban and rural residents [12]. Our findings coincide with the research results of some scholars: Considering the time-varying relationship, there is a dynamic inverted U-shaped between financial development and the urban-rural income gap, and different regions are in different stages of the inverted U-shaped curve [14]. Analyzing the direction and effect of China's economic financialization on the urban-rural income gap from both the national and regional

**Table 6. FMOLS DOLS CCR long term estimates.**

| Dependent variable: lnGI | | | | | | |
|---|---|---|---|---|---|---|
| Variables | Model (1) | | | Model (2) | | |
| | FMOLS | DOLS | CCR | FMOLS | DOLS | CCR |
| C | 5.0290*** | 5.0152*** | 4.7773*** | 2.8399*** | 2.8478*** | 2.7111*** |
| | [0.0036] | [0.0015] | [0.0051] | [0.0001] | [0.0028] | [0.0001] |
| lnTI | 0.1380*** | 0.1364*** | 0.1279*** | 0.1387*** | 0.1384*** | 0.1272*** |
| | [0.0002] | [0.0020] | [0.0000] | [0.0001] | [0.0035] | [0.0000] |
| lnFIR | 0.9423*** | 0.9377** | 0.8817*** | - | - | - |
| | [0.0051] | [0.0445] | [0.0065] | - | - | - |
| lnFIR$^2$ | - | - | - | -0.1013*** | -0.1015** | -0.0938*** |
| | - | - | - | [0.0035] | [0.0316] | [0.0046] |

Note: The p-values are given in [].

***, ** and denote significance at 1% and 5% levels, respectively.

levels, it is settled that there is a Kuznets inverted U-shaped trajectory between the financial development of the western region and the urban-rural income gap, and it is directly in the stage of diminishing the income gap [38].

The long-term estimates of Models (1) and (2) obtained from the FMOLS, DOLS and CCR methods indicate that the coefficient of technological innovation is positive and statistically significant at the significance level of 1%. This reveals that technological innovation increases income gap in China. This may be due to the following reasons. The skill-biased type of technological progress increases the remuneration of high- and low-skilled workers, resulting in polarization and widening the income gap. The result of the skill spillover effect will eventually lead to wage differences among workers with the same skill level. Moreover, Technological innovation may hinder small and medium-sized enterprises from entering monopoly and oligopoly industries. The wage gap between workers with different skill levels is determined by the relative supply of skilled workers and skill spillover efficiency among workers with different skill levels. Our empirical finding is consistent with Aghion et al., who finds that technological innovation has a positive impact on income inequality in the US [39]. Likewise, Mnif reveals that technological innovation exacerbates income inequality in 19 developing countries [40].

## VECM Granger causality test

On the basis of the above test results, in the case of the optimal lag order of order 1, this section will continue to test the long-term and short-term causality of ΔlnGI, ΔlnTI and ΔlnFIR based on VECM. This method verifies the causality between variables in the composite system, which can avoid the disadvantage that the traditional Granger causality test can not be applied to the cointegration test. The causality results presented in Table 7 suggest that financial development and income inequality cause each other. This conclusion is the same as that of Zhang Yingli and Yang Zhengyong [41]. He uses the VECM model to dynamically analyze the relationship between financial development, urbanization, and the urban-rural income gap. The empirical results show that financial development is the uni-directional Granger cause of the urban-rural income gap. The causal relationship results also show that there is a bidirectional causal linkage between technological innovation and urban-rural income gap at the significance levels of 1% and 5%. Ma Lei explored the impact of human capital structure and total factor productivity on urban-rural income gap from the perspective of innovation-driven

**Table 7. VECM Granger causality test.**

| Dependent variable | Independent variable Short-run | | | Long-run [p-value] |
|---|---|---|---|---|
| | F-statistic [p-value] | | | |
| | ΔlnGI | ΔlnTI | ΔlnFIR | ECTt-1 |
| ΔlnGI | - | 13.824*** | 9.4808** | -0.0003** |
| | | [0.0008] | [0.0044] | [0.035] |
| ΔlnTI | 7.4338** | - | 7.5969*** | -0.0015** |
| | [0.0107] | | [0.0098] | [0.006] |
| ΔlnFIR | 5.0151** | 4.2866** | - | -0.0010*** |
| | [0.0327] | [0.0471] | | [0.000] |

Note:

*** and **denote significance at 1% and 5% levels, respectively.

development [27]. However, there is no research on the causal relationship between technological innovation and urban-rural income gap.

Overall, short-term fluctuations and long-term equilibrium characterize the relationship between financial development, technological innovation, and the urban-rural income gap. Financial development and technological innovation will have an impact on the urban-rural income gap, which is the result of China's financial development bias, and it is also an inevitable phenomenon that the process of technological innovation and development has an impact on the income distribution pattern of different groups.

The bias in financial development has had an impact on the expansion of the urban-rural income gap in China to a certain extent. On the one hand, the profit orientation of capital and China's financial policies focus on supporting urbanization, resulting in a part of rural funds entering the urban financial market, accelerating the economic development of cities and the increase of urban residents' income, but not affecting rural construction, hindering the development and growth of the rural economy. On the other hand, in the case of rural finance lagging behind urban financial development, due to the imperfect rural financial market and mechanism, the level of rural investment and consumption is low, which is not conducive to the sustainable development of the cause of agriculture, rural areas and farmers and reduces the income of rural residents. Coupled with the government's rural financial support for agricultural development being relatively weak, the serious degradation of financial institutions to support agriculture, financial institutions in rural areas 'retreating weaker and weaker, weaker and retreating' phenomenon, leading to the rural financial market into a small scale, low-efficiency development dilemma. These long-term constraints have made it very difficult to narrow the income gap between urban and rural residents and gradually separated from poverty alleviation and rural revitalization.

The different effects of financial development on state-owned enterprises and private enterprises are also closely related to the urban-rural income gap. Since the reform and opening up, China has adopted 'gradual reform', and one of the important strategies is the dual track reform strategy, that is, plan and market, state-owned enterprises and private enterprises coexist. In the early days of reform and opening up, the government took supporting the development of state-owned enterprises as its primary goal, and also actively provided policy support for the development of small and medium-sized enterprises and private enterprises. However, with the deepening of the reform, the problem of low operating efficiency of state-owned

enterprises has gradually emerged. To support the continued operation of state-owned enterprises, government departments have provided disguised subsidies to state-owned enterprises in various ways. The bias of financial policies toward state-owned enterprises will inevitably lead to more difficult living spaces for small and micro enterprises or private enterprises. This shows that the services currently provided by China's financial industry cannot fully meet the financing needs of private enterprises and small and medium-sized enterprises, and the financial system still has obvious shortcomings. Although the dual-track reform is intended to support the common development of state-owned enterprises and private enterprises, the actual implementation of the financial policy still prefers state-owned enterprises, and SMEs can not be treated equally. Small and micro enterprises and private enterprises are difficult to promote their development by financial means, which is not conducive to alleviating the income gap.

Due to the 'urban-rural dual' structure of the national industrial policy, there are differences in scientific and technological innovation ability and innovation efficiency between urban and rural areas.

Agricultural scientific and technological innovation does not match human capital, the accumulation rate of agricultural high-quality human capital lags behind the needs of technological innovation, and agricultural technological innovation has a weak impact on the urban-rural income gap; The urban industrial sector and science and technology service sector have an enormous investment in innovation resources, high scientific and technological innovation ability, high innovation efficiency, large profit space, and rapid production efficiency improvement. In addition, the dual economic structure of urban and rural causes the endowment of household resources and the education level of farmers less than in the city, which affects the increase of rural residents' income and increases the income gap between urban and rural residents.

## Conclusion and policy suggestion

There is no relevant research on the relationship between financial development, technological innovation and urban-rural income gap in the existing literature. In the current environment of steady economic development and building a harmonious society, narrowing the urban-rural income gap is an important problem in the process of economic development in China. Therefore, this study investigates the FKC for urban-rural income gap in case of China for the period of 1985–2019. This study has intensified on the technological innovation-income gap link with the FKC. For this purpose, we apply the ARDL approach and Johansen method for cointegration. In addition, the long run coefficient estimates are conducted by DOLS, FMOLS and CCR estimators. We also apply the VECM Granger procedure to causality. Finally, we using OLS regression analysis to variables. Compared with existing research, this study has made improvements in the following two aspects: This paper will add the indicator of technological innovation. On the one hand, there is no literature on the relationship between the three in China. On the other hand, technological innovation is closely related to financial development and the income gap between urban and rural residents. Thus, the level of technological innovation cannot be abandoned in the study; It will verify the FKC theory and analyze the link between technological innovation and urban-rural income gap under the FKC hypothesis.

It is found that the long-run relationship exists among the variables under the structural breaks. The main finding obtained from the long-run coefficient estimates reveal that technological innovation increases urban-rural income gap. The findings confirm the validity of the FKC hypothesis for China's economy in the long run. The causality analysis shows a bi-directional causality between financial development, technological innovation and urban-rural

income gap in the long run. We can present the following policy suggestions for China's economy.

First, expand the scale of rural financial development and improve the efficiency of rural financial development. Because of the difficulty of loans in rural areas and the outflow of rural funds from rural areas through household savings and savings of township enterprises, when farmers have financing needs, they cannot get corresponding financing support because they cannot meet the credit threshold of financial institutions, and the efficiency of rural financial allocation is low. Therefore, the government should establish a sound agricultural financial system, optimize the share of agricultural financial services and financial resources, and try to improve the uneven distribution of financial resources between urban and rural areas and financial development to benefit more high-income people and less rural residents. Increase preferential policy support for rural economic development, improve the rural financial organization system, and promote agricultural modernization. In addition, the continuous expansion and upgrading of rural agricultural financial services will help attract more financial institutions and financial products to enter the countryside, and will also help rural areas reduce financing costs and thresholds, and reduce the loss of rural financial resources and talents. The improvement of rural financial operation efficiency can convert the absorbed rural savings into rural loans in real time, increase support for 'agriculture, rural areas and farmers' funds, increase the utilization rate of rural financial resources, and improve the development of rural financial development.

Second, Financial regulators should appropriately relax the market access threshold, encourage rural commercial banks, small and medium-sized commercial banks, financing guarantee companies, and other institutions to take root in the countryside, form a diversified rural financial service system that is competitive and coexisting with each other, and constantly improve the financial availability of farmers.

Third, adjust the structure of financial development, improve the efficiency of financial development, and narrow the income gap between urban and rural residents. We should constantly improve the financial market, establish a perfect and multi-level financial system that adapts to the development of the times, and set up corresponding regulatory authorities to give full play to the independent effect of the financial market. In terms of the income distribution, we should not only focus on efficiency but also fairness in distribution. So should financial development. We should pay attention to the efficiency and structure of financial development while expanding the scale of financial development. The focus of financial development before was mainly on expanding the scale, and always ignored the efficiency of financial development. Therefore, while developing finance, we should take into account the expansion of development scale and efficiency improvement, but we can not ignore the rationality of financial development structure.

Last, the government should take practical measures and introduce preferential policies to support technological innovation of small and medium-sized enterprises, create good financing conditions for SMEs, and establish and develop technological service systems for SMEs. In addition, local governments should insist on promoting industry to agriculture in technological innovation, cities supporting rural areas, and continuously increasing the intensity of financial investment in science and technology. On the one hand, while creating a good background to promote technological innovation, it should also introduce and implement relevant policies to promote technological innovation, create special innovation funds, actively subsidize and encourage scientific and technological innovation in large, medium, and small enterprises, research and development institutions, universities and other major scientific and technological innovation achievements. Vigorously promote industry-university-research cooperation between enterprises and universities, and help enterprises expand the market for

technological innovation products and services. On the other hand, actively establish and improve the agricultural technology innovation coordination mechanism, improve the construction of agricultural technology innovation software and hardware, raise the level of agricultural technology, and lay a solid foundation for the construction of new rural industries.

This paper has some limitations. First, since we strengthen the relationship between technological innovation and urban-rural income gap in the context of FKC hypothesis, several explanatory variables such as globalization, human capital, and renewable energy are not included in our specifications. Second, many indicators of financial developments such as current liabilities and financial development index are not used for empirical analysis. At the same time, the study may inspire future researches. In this paper, there are some limitations in the relationship between patent licensing data analysis and urban-rural income gap. In the follow-up study, independent technological innovation indicators such as high-tech product export and R&D investment can be considered to further demonstrate the impact of technological innovation on urban-rural income gap and the differences in the results of different indicators. Therefore, future researches can investigate the impact of technological innovation on urban-rural income gap in detail and prefer comparative empirical results.

## Supporting information

**S1 Table. Urban-rural income gap, financial development level and technological innovation level in China, 1985–2019.**
(XLS)

## Author Contributions

**Conceptualization:** Li-Min Wang.

**Data curation:** Li-Min Wang.

**Formal analysis:** Li-Min Wang.

**Funding acquisition:** Xiang-Li Wu.

**Investigation:** Li-Min Wang.

**Methodology:** Li-Min Wang.

**Project administration:** Li-Min Wang.

**Resources:** Li-Min Wang, Xiang-Li Wu, Nan-Chen Chu.

**Software:** Li-Min Wang, Xiang-Li Wu.

**Supervision:** Li-Min Wang.

**Validation:** Li-Min Wang.

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
