## [Decision Letter · Decision Letter 0]

10 Jun 2022

PONE-D-22-09786Financial development, technological innovation and urban-rural income gap: time series evidence from ChinaPLOS ONE

Dear Dr. Wu,

Thank you for submitting your manuscript to PLOS ONE. After careful consideration, we feel that it has merit but does not fully meet PLOS ONE’s publication criteria as it currently stands. Therefore, we invite you to submit a revised version of the manuscript that addresses the points raised during the review process.

We look forward to receiving your revised manuscript.

Kind regards,

Ming Zhang, Ph.D.

Academic Editor

PLOS ONE

Journal Requirements:

[The authors declare that they have no known competing financial interests or personal relationships that could have appeared to influence the work reported in this paper.]

3. We note you have included a table to which you do not refer in the text of your manuscript. Please ensure that you refer to Table 2 in your text; if accepted, production will need this reference to link the reader to the Table.

Reviewers' comments:

Reviewer's Responses to Questions

**Comments to the Author**

1. Is the manuscript technically sound, and do the data support the conclusions?

Reviewer #1: Partly

Reviewer #2: Yes

2. Has the statistical analysis been performed appropriately and rigorously? 

Reviewer #1: Yes

Reviewer #2: Yes

3. Have the authors made all data underlying the findings in their manuscript fully available?

Reviewer #1: Yes

Reviewer #2: Yes

4. Is the manuscript presented in an intelligible fashion and written in standard English?

Reviewer #1: Yes

Reviewer #2: Yes

5. Review Comments to the Author

Reviewer #1: 1. The Introduction fails to motivate the study. In the present form, it resembles a mini-review of literature, rather than discussing any policy-level problem. Why this study is necessary? What policy level problem this study is addressing? How the study is expected to provide any solution to that problem? How does the choice of sample is complementing that problem? Are the results and policies generalizable? The introduction is silent in all these aspects. The mere choice of new variables, new methods, or choosing a new context is not considered as contribution of a study.

Try to find out the policy level problem persisting in the chosen countries from the third-party research reports, as academic literature will not be able to provide you with the specific policy issue. Limit the number of references in the Introduction to a maximum of 6.

2. What is the aim of the review of literature? The authors have merely listed out the studies without even creating a debate among them. Without that debate and thoughtful contradictions, the research gap cannot be substantiated.

3. How the authors have derived the empirical model? There should be a thorough theoretical underpinning behind the model. This section should be based on the logic of the authors, and no citation/reference should appear here. This section will be followed by the empirical model.

4. Authors have merely reported the results without even discussing the economic intuitions behind the results. Are these results supporting or refuting the existing policies in the chosen context? Are the results directed towards any new policy initiatives? The discussion of results should open up the threads of policy discussion, which is completely absent in this case. A mere comparison of the results with the literature doesn't ensure the novelty of the results unless they give out something new on the theory/policy front.

5. Conclusion reiterates the results, which is completely undesirable. The authors should summarize the results within a maximum of 3 sentences. Moreover, the policies are completely vague, and it seems that the authors already had the policies in mind before even starting the paper. The policies should be directly derived from the discussion of the results, and they should not go beyond the results.

Reviewer #2: The study is well designed. It provides several contrubitions to the literature. Empirical methods are well selected to develop a model related to urban-rural income gap. Policy suggestions are enough. However, the following suggestions should be carried out for improving the study:

1. The results of causality analysis should be discussed in detailed with the findigs of other empirical studies.

2. To improve the empirical litarature, the following references should be integrated into the literature:

i) Financial Development, Technological Innovation and Income Inequality: Time Series Evidence from Turkey. Soc Indic Res 156, 47–69 (2021). https://doi.org/10.1007/s11205-021-02641-7

ii) Income inequality and CO2 emissions: nonlinear evidence from Turkey. Environ Dev Sustain (2021). https://doi.org/10.1007/s10668-021-01922-y.

iii) The impact of financial development on income inequality: An empirical evidence for Turkish economy. International Anatolia Academic Online Journal Social Sciences Journal, 2015, 3(2), 52-63.

6. PLOS authors have the option to publish the peer review history of their article (what does this mean?). If published, this will include your full peer review and any attached files.

Reviewer #1: No

Reviewer #2: No

---

## [Author Response · Author response to Decision Letter 0]

8 Jul 2022

Thank you for your letter and for the reviewers' and editors' comments concerning our manuscript. These comments are very helpful for revising and improving our paper. We have studied comments carefully and have made corrections which we hope meet with approval.

Reviewer #1comments1: 

The Introduction fails to motivate the study. In the present form, it resembles a mini-review of literature, rather than discussing any policy-level problem. Why this study is necessary? What policy level problem this study is addressing? How the study is expected to provide any solution to that problem? How does the choice of sample is complementing that problem? Are the results and policies generalizable? The introduction is silent in all these aspects. The mere choice of new variables, new methods, or choosing a new context is not considered as contribution of a study. 

Response: Thanks for your valuable comments and suggestions. We have revised and supplemented the introduction, including the necessity of this study, the policy issues to be solved by the study, and the reasons for the selection of research samples. To stimulate research. At the same time, the academic literature cited in the introduction has been greatly deleted to find out the long-standing policy problems in China. The details are as follows:

Income distribution has always been a popular issue of the entire society. Reasonable income distribution system and income gap are not only the common aspiration of the people but also an important embodiment of social justice. Since the reform and opening up, China's economy has achieved sustained and rapid development, and the living standards of residents have gradually improved. However, while per capita income is flourishing, the imbalance in economic development is also increasing. At present, there are still four major gaps: regional gaps, urban-rural gaps, industry gaps, and class gaps. Among them, the income gap between urban and rural residents has attracted the wide attention of the academic community, because it has the largest proportion in the causes of the overall income gap of Chinese residents, up to over 65%. Especially in recent years, with the widening gap between urban and rural residents' income growth rate, the income gap between urban and rural areas in China continues to increase, which has become an obstacle to the long-term healthy and stable development of China's economy. The data from China Statistical Bureau show that the income ratio of urban and rural residents in China has decreased from 3.30: 1 in 2007 to 2.64: 1 in 2019, and the overall trend is declining. However, compared with the world level, the income gap between urban and rural areas in China is hovering at a high level and shows a fluctuating trend. During 2002-2019, it expanded at a high level before 2009, and then fell within a narrow range, showing an inverted U-shaped trend. At this stage, the income distribution of residents in China is still very serious, the income gap between urban and rural residents fluctuated and expand, which will inevitably affect social harmony and stability. Therefore, in realizing the path of establishing and improving the institutional mechanism of urban-rural integration development and balancing urban-rural integration development put forward in the 19th National Congress of the Communist Party of China, solving the urban-rural income gap has become the key point the sustainable economic development at this stage and in the future for a long time.

Various comprehensive factors affect the income gap between urban and rural areas, including the national fiscal policies and the national financial development, especially the reason for widening the gap is the imbalance of urban and rural financial development. For example, when financial resources do not support rural residents’ income increase, the urban-rural income gap will continue to expand. If the urban-rural income gap continues to expand, it will strengthen the imbalance between urban-rural financial resource allocation, which will lead to a vicious cycle of imbalance. Since the establishment of agriculture-related loan statistics in 2007, the amount of agriculture-related loans of national financial institutions has increased by 361.7%, with an average annual growth rate of 18.8% in ten years. The amount of agriculture-related loans increased from 6.1 trillion yuan at the end of 2007 to 30.95 trillion yuan at the end of 2017, an increase of 4.5 percent in the proportion of loans. Rural loans and savings have increased substantially. However, China’s rural financial supply still cannot keep up with the pace of rapid growth in demand, the financial correlation rate in rural areas, the proportion of deposits and loans on the financial development scale, and the efficiency of financial institutions are still lower than those in cities. In addition, a high monopoly characterizes China’s current financial structure. Compared with cities, corporate loans in rural areas are more disadvantaged, resulting in more scarce rural financial resources. According to statistics, there is still a gap in financial services in over 1200 townships in China. Over 8,000 townships only rely on the only financial organization to carry out basic financial business. The coverage rate of rural loans across the country does not exceed 35%. Therefore, China's financial development presents a typical 'dual structure. Technological innovation is considered to be the main factor affecting macroeconomic variables. Hou Zhenmei, Tian Mao, et al. [1] pointed out that the process of technological innovation is an economic growth process that takes technology as the mainline and causes social change and technological innovation is a key driving force for economic growth, and its development process will inevitably affect the change of income distribution pattern of different groups. Pan et al. [2], Wang and Wang [3] dwell on the influence of technological innovation on energy efficiency. Technological innovation can influence environmental degradation [4]. Recently, the effect of innovation on the income gap is an important field of research [5-6]. 

The China's economy is an essential case. In 2016, China became the first middle-income economy to enter the top 25 of the global innovation index. China's scientific and technological innovation capability was enhanced and the main scientific and technological innovation indicators were steadily improved in 2018. The state intellectual property office of the data shows the proportion of research and experimental development expenditure of the entire society in GDP in 2018 was 2.15%. The total number of R&D personnel reached 4.18 million person-years, ranking first in the world; The number of international scientific papers and citations ranked second in the world; The number of patent applications and authorization of invention ranks first in the world; The contribution rate of scientific and technological progress is expected to exceed 58.5%, and the national comprehensive innovation ability ranks 17th in the world. The level of scientific and technological innovation in China has been continuously improved, and fruitful achievements have been made in many fields such as innovation input, innovation output, and innovation efficiency. These developments indicate technological innovations advance rapidly in China. In addition, income distribution has been a vital matter in China for a long time. According to data released by the National Bureau of Statistics, China's urban-rural income gap in 2019 was 2.7:1, higher than the world average, and is one country with the largest urban-rural income gap. In this context, to incorporate technological innovation into the important factors that affect the income gap between urban and rural areas, and to explore ways and channels to ease and eliminate the excessive income gap in residential areas in China, which is of great significance to reduce income inequality and realize social equity at this stage. 

Considering all the above explanations, the present study creates the following questions: First, is the FKC hypothesis valid for China? Second, how technological innovation influences income gap in China's economy? Third, is there a causal linkage between technological innovation and urban-rural income gap? In addition, there are no studies on the causal linkage between them. In this regard, this is the first study on China's economy. To answer these questions above, we attempt to make an empirical study on the link between technological innovation and income gap in China from 1985 to 2019 under the context of FKC hypothesis. We intensify on the long-term and causal relations between the variables. In the study of financial development and the income gap between urban and rural residents, this paper adds the index of technological innovation. By citing these three variables, hope to be more comprehensive on the causes of China 's urban-rural income gap and how to solve this problem are discussed, and thus draw more accurate results and set forward more constructive suggestions. The widening income gap between urban and rural areas in China has attracted wide attention from the government and academia, but from the domestic existing literature, it focuses on the role of law decision-making factors. There is no analysis on the impact of technological innovation on the income gap under the background of FKC hypothesis. In addition, the utility research on the long- and short-term causality test between technological innovation and income gap is indeed more inadequate. Thus, the study intends to focus on verifying the validity of the FKC hypothesis and the dynamic relationship between technological innovation and the urban-rural income gap in the FKC hypothesis. Producing a new explanation for the causes of urban-rural income gap, this has important theoretical and practical relevance for developing regional coordinated expansion and achieving social equity.

Thanks! 

Reviewer #1comments2: 

What is the aim of the review of literature? The authors have merely listed out the studies without even creating a debate among them. Without that debate and thoughtful contradictions, the research gap cannot be substantiated.

Response: Thanks for your valuable comments and suggestions. We have supplemented the domestic and foreign literature on income gap, and created a debate among them. The whole literature review has been revised, supplemented and sorted out. The details are as follows:

The urbanization process and financial development of developed countries began earlier. Foreign scholars have carried out relatively full research on financial development and income gap, but no consensus has been reached and there is a big controversy. FKC Hypothesis was first developed by Kuznets [7], an American economist and statistician, through an analysis of data related to economic growth and income gap in several countries, there is an inverted U-shaped relationship between financial development and urban-rural income gap. This is because, in the early stage of financial development, only a few high-income people can pay the cost of financial services and get high returns financial services, so the income gap expands; When the financial development is to a certain extent, most people can cross the wealth threshold of financial services, share the benefits of financial development, the income gap narrowed slowly. Then Greenwood and Jovanovic [8] found developed a GJ model to explore the communication between financial development and economic growth and income distribution and found that they also conformed to the 'inverted U' trend. Townsend and Ueda improved the model on this basis and reached similar conclusions [9]. However, Clarke [10] adjusted the research data to the panel data of several countries and found that from an international perspective, financial development and income gap did not show an 'inverted U' feature, but were negatively correlated. Further deepening of financial development would have a positive effect on narrowing the urban-rural income gap. Pradhan [11] used time-series data from India 45 to show the relationship between economic growth, financial development, and the income gap between urban and rural residents, and also found that financial development is conducive to narrowing the income gap. However, some scholars have reached different conclusions in their studies. Law et al. [12] used the threshold regression method to conduct an empirical analysis of financial development and income gap. The results show that there is a threshold effect between them. Only after reaching a certain threshold, financial development will narrow the income gap, and the implementation does not exist before. Sehrawat and Giri use ARDL model analysis results show that both in the long and short term. India's financial development level has increased the income gap [13].

Since the 1990s, with the swift growth of financial markets and the advance of urbanization, the class bias of communal income has come better and further serious. Domestic scholars analyzed and discuss the issue of the urban-rural income gap. Many scholars have different perspectives on the impact of financial development on the urban-rural income gap. For example, Qiao Haishu and Chen Li [14], Hu Zongyi and Liu Yiwen [15], and Yang Nan and Ma Chuoxin [16] upheld that the impact mechanism of China's financial development on the urban-rural income gap is also in line with the 'inverted U' component. Based on the theory of economic growth, Ye Zhiqiang et al. [17] analyzed and discussed the impact of financial development on narrowing the income gap, and thought that the problem of poverty in rural China cannot be solved effectively by developing finance alone, and unreasonable financial expansion will lead to the widening of the income gap between urban and rural areas. Jia Fei found that the low financial transformation efficiency in rural areas will lead to the imbalance of urban and rural financial development and further aggravates the widening income gap [18]. From the standpoint of the regional economy, Sun Yongqiang et al. [19] pointed out that for a long time, China's financial development and financial development showed a significant positive correlation, especially in the eastern region. Yang Youcai used the threshold model to verify the threshold effect of financial development and found that the financial threshold of China's regions showed an increasing trend from west to east. However, due to the differences between the corresponding thresholds of the level of financial development in the eastern and western regions, there are also differences in the changes in the income gap caused by financial growth in the eastern and western regions [20].

Existing studies have shown that technological innovation is a key determinant of the urban-rural income gap. Technological innovation promotes economic development and pushes the development process of industrialization and urbanization. What is the impact of technological innovation on the income gap? Foreign scholars have conducted in-depth research on the impact of technological innovation on income disparity and the results of the study are also different. The major research results focus on the following aspects:

Firstly, the skill bias of technological progress makes the pay growth of high-skilled and low-skilled workers polarization, which leads to the widening income gap, the effect of skill-biased technological progress. Leamer believes that technological progress in technology-intensive sectors contributes to higher wages for skilled workers, in labour-demanding parts contributes to higher wages for unskilled workers, and that technological progress in technology-intensive sectors has a more pronounced impact on income gaps [21].

Second is the spillover effect of skills. Aghion and others found new technologies can improve labor productivity, and workers can learn to increase their knowledge [22]. The result is that the wages of workers in the same skill level group are rare, and the relative supply and technology spillover efficiency of skilled workers among workers various skill levels cause the wage gap between workers with different skill levels. Cirillo.V analysis shows that technological innovation is conducive to narrowing the overall income gap, but it will aggravate the income gap among high-income groups [23].

The third is the Organizational Effect on Labor Market. Nathan M points out that technological innovation affects the direction and intensity of the income gap, which is also related to aspects such as the size of the labor force, the structure of labor skills and the degree of economic development [24]. Akcigit et al. [25] remarked that innovation exacerbates the gap between high-income meets and that the income of most successful innovators has risen dramatically.

Domestic scholars have also studied this issue from multiple perspectives, aiming at the relationship between technological innovation and income gap. Some studies believe that technological innovation tends to widen the income gap. Chen Yong and Bai Zhe found that skill-biased technological progress is the most important factor in the widening regional wage gap in China [26]. Zeng Peng et al. [27] found that in terms of China’s urban agglomerations, technological progress will expand the urban-rural income gap, and technological progress will promote the improvement of urbanization. However, some scholars believe that technological innovation can alleviate the income gap. Dong Zhiqing et al. [28] and others believe that neutral technological advances can raise the supply of skilled labour and reduce the wage gap between skilled and non-skilled labour. Ma Lei discussed the impact of total factor productivity and human capital structure on urban-rural income gap from the perspective of innovation-driven development [29]. The study found that the optimization of human capital structure in the central and western regions has a powerful effect on reducing the urban-rural income gap. The growth of total factor productivity and the improvement of technological progress can narrow the gap in the central and western regions, but it shows an expansion effect in the eastern field.

From the above summary, we can see that the existing literature has conducted in-depth research on the problem of urban-rural income gap, and applied research has also made abundant achievements. However, there are still some shortcomings which need to be further studied: Firstly, the empirical results related to FKC hypothesis are very complex and inconsistent. The main reason for this statement is that periods, analysis techniques and country groups considered are different. Secondly, most of the studies focus on the validity of FKC hypothesis, but they do not integrate technological innovation into the income gap specifications. Lastly, even if several studies only investigate the technological innovation-income gap link, they do not dwell on the FKC hypothesis. The research on the above three points can be regarded as the contribution of this paper. Therefore, the present study aims at examining the relationship between technological innovation and urban-rural income gap in the presence of FKC hypothesis for China’s economy, and put forward policy recommendations on this basis.

Thanks!

Reviewer #1comments3:

How the authors have derived the empirical model? There should be a thorough theoretical underpinning behind the model. This section should be based on the logic of the authors, and no citation/reference should appear here. This section will be followed by the empirical model.

Response: Thanks for your valuable comments and suggestions. We have supplemented the theoretical underpinning behind the model and removed the cited citation/reference. The details are as follows:

Materials and method

The selection of variables

Technological innovation. The technological innovation process of a country is embodied in the economic growth process that takes technology as the main line and causes social changes. The process of technological innovation and development will inevitably affect the changes of income distribution pattern of different groups. In measuring the level of technological innovation, there are usually two interrelated patent indicators, namely the amount of patent authorization and the amount of patent acceptance, which are widely used in the literature. Compared with the patent acceptance, the index of patent authorization can more calculate the innovation level. Therefore, this paper will still use this index to measure the level of technological innovation in China.

Financial development. Goldsmith proposed that the financial correlation ratio refers to the ratio of the value of all financial assets of a country to the total amount of economic activities in that country at a certain date. The change in financial related ratio reflects the changing relationship between the financial sector and the economic base in terms of scale, which is used to measure the scale of a country's or region's financial development and the degree of financial deepening. Based on the fact that China's financial market is not perfect and the scale of the banking industry accounts for a high proportion of the financial industry, So this study selects the Gohren index (FIR) which is measured by the ratio of the end-of-year loans of financial institutes to GDP. Since the inverted U-shaped contact between the level of financial development and the income gap has regularly been a hot subject in academic research. Scholars all use this variable as an important factor in analyzing income disparity. Therefore, this paper introduces the financial development scale (FIR) and its quadratic call to demonstrate whether there is an inverted U-shaped link between the two.

Urban-rural income gap. Indicators to measure the income gap between urban and rural residents are Gini coefficient, Theil coefficient, Wolfson polarization index, etc. Based on the pertinent indexes selected in this article need to reflect the efficiency of the input and output level of urban and rural residents' productivity, so this paper selects the urban residents' per capita disposable income UI and the ratio of rural residents' per capita net income RI to measure the income gap between urban and rural residents.

Thanks!

Reviewer #1comments4:

Authors have merely reported the results without even discussing the economic intuitions behind the results. Are these results supporting or refuting the existing policies in the chosen context? Are the results directed towards any new policy initiatives? The discussion of results should open up the threads of policy discussion, which is completely absent in this case. A mere comparison of the results with the literature doesn't ensure the novelty of the results unless they give out something new on the theory/policy front.

Response: Thanks for your valuable comments and suggestions. We have added economic intuitions and policy discussion behind the results. The details are as follows:

Estimation Results

After checking the co-integration between the series, the long-term parameters can be estimated by the FMOLS, DOLS and CCR estimators. The estimation results are shown in Table 6. The coefficients of lnFIR and lnFIR2 are found positive and negative at 1% level of significance, respectively. The findings are consistent with the validity of the FKC hypothesis proposed by Greenwood and Jovanovich [8]. Thus, this reveals that there exists an inverted-U shaped relationship between financial development and urban-rural income gap in China’s economy. Namely, in the initial stage of financial development, the income gap between urban and rural areas continues to expand. When developed to a certain extent, because of the diffusion and infiltration of financial resources from central cities to surrounding rural areas, and improve the financial imbalance. The scale of rural financial resources is expanded and the utilization efficiency is raised. Capital accumulation and technological progress in rural areas promote sustained income growth of rural residents, then narrow the income gap between urban and rural residents [14]. Our findings coincide with the research results of some scholars: Considering the time-varying relationship, there is a dynamic inverted U-shaped between financial development and urban-rural income gap, and different regions are in different stages of the inverted U-shaped curve [16]. Analyzing the direction and effect of China’s economic financialization on the urban-rural income gap from both the national and regional levels, it is settled that there is a Kuznets inverted U-shaped trajectory between the financial development of the western region and the urban-rural income gap, and it is directly in the stage of diminishing the income gap [40].

The long-term estimates of Models (1) and (2) obtained from the FMOLS, DOLS and CCR methods indicate that the coefficient of technological innovation is positive and statistically significant at the significance level of 1 %. This reveals that technological innovation increases income gap in China. This may be due to the following reasons. The skill-biased type of technological progress increases the remuneration of high- and low-skilled workers, resulting in polarization and widening the income gap. The result of the skill spillover effect will eventually lead to wage differences among workers with the same skill level. Moreover, Technological innovation may hinder small and medium-sized enterprises from entering monopoly and oligopoly industries. The wage gap between workers with different skill levels is determined by the relative supply of skilled workers and skill spillover efficiency among workers with different skill levels. Our empirical finding is consistent with Aghion et al., who finds that technological innovation has a positive impact on income inequality in the US [41]. Likewise, Mnif reveals that technological innovation exacerbates income inequality in 19 developing countries [42].

Table 6. FMOLS DOLS CCR long term estimates

Dependent variable：lnGI 

Variables Model（1） Model（2） 

 FMOLS DOLS CCR FMOLS DOLS CCR

C 5.0290***

[0.0036] 5.0152***

[0.0015] 4.7773***

[0.0051] 2.8399*** 2.8478*** 2.7111***

 [0.0001] [0.0028] [0.0001]

lnTI 0.1380***

[0.0002] 0.1364***

[0.0020] 0.1279***

[0.0000] 0.1387*** 0.1384*** 0.1272***

 [0.0001] [0.0035] [0.0000]

lnFIR 0.9423***

[0.0051] 0.9377**

[0.0445] 0.8817***

[0.0065] - - -

 - - -

lnFIR2 -

- -

- -

- -0.1013***

[0.0035] -0.1015**

[0.0316] -0.0938***

[0.0046]

 Note: The p-values are given in []. ***, ** and denote significance at 1% and 5% leves,

 respectively.

VECM Granger causality test

On the basis of the above test results, in the case of the optimal lag order of order 1, this section will continue to test the long-term and short-term causality of △lnGI、△lnTI and △lnFIR based on VECM. This method verifies the causality between variables in the composite system, which can avoid the disadvantage that the traditional Granger causality test cannot be applied to the cointegration test. The causality results presented in Table 7 suggest that financial development and income inequality cause each other. This conclusion is the same as that of Zhang Yingli and Yang Zhengyong [43]. He uses the VECM model to dynamically analyze the relationship between financial development, urbanization, and the urban-rural income gap. The empirical results show that financial development is the uni-directional Granger cause of the urban-rural income gap. The causal relationship results also show that there is a bidirectional causal linkage between technological innovation and urban-rural income gap at the significance levels of 1% and 5%. Ma Lei explored the impact of human capital structure and total factor productivity on urban-rural income gap from the perspective of innovation-driven development [29]. However, there is no research on the causal relationship between technological innovation and urban-rural income gap.

Table 7. VECM Granger causality test

Dependent variable Independent variable

Short-run Long-run

[p-value]

 F-statistic

[p-value] 

 △lnGI △lnTI △lnFIR ECTt-1

△lnGI ¬- 13.824***

[0.0008] 9.4808**

[0.0044] -0.0003**

[0.035]

△lnTI 7.4338**

[0.0107] - 7.5969***

[0.0098] -0.0015**

[0.006]

△lnFIR 5.0151**

[0.0327] 4.2866**

[0.0471] - -0.0010***

[0.000]

Note: *** and **denote significance at 1% and 5% levels, respectively.

Overall, short-term fluctuations and long-term equilibrium characterize the relationship between financial development, technological innovation, and the urban-rural income gap. Financial development and technological innovation will have an impact on the urban-rural income gap, which is the result of China's financial development bias, and it is also an inevitable phenomenon that the process of technological innovation and development has an impact on the income distribution pattern of different groups. 

The bias in financial development has had an impact on the expansion of the urban-rural income gap in China to a certain extent. On the one hand, the profit orientation of capital and China's financial policies focus on supporting urbanization, resulting in a part of rural funds entering the urban financial market, accelerating the economic development of cities and the increase of urban residents' income, but not affecting rural construction, hindering the development and growth of the rural economy. On the other hand, when rural finance lags behind the development of urban finance, due to the imperfect rural financial market and mechanism, the low level of rural investment and consumption, coupled with the relatively weak government support for rural finance to agricultural development, the agricultural support function of financial institutions has been seriously degraded, resulting in the phenomenon that financial institutions are 'retreating weaker and weaker, weaker and retreating' in rural areas, leading to the development dilemma of small-scale and low efficiency of rural financial markets, these long-term constraints lead to narrowing the income gap between urban and rural residents has become quite difficult.

Due to the 'urban-rural dual' structure of the national industrial policy, there are differences in scientific and technological innovation ability and innovation efficiency between urban and rural areas.

Agricultural scientific and technological innovation does not match human capital, the accumulation rate of agricultural high-quality human capital lags behind the needs of technological innovation, and agricultural technological innovation has a weak impact on the urban-rural income gap; The urban industrial sector and science and technology service sector have an enormous investment in innovation resources, high scientific and technological innovation ability, high innovation efficiency, large profit space, and rapid production efficiency improvement. In addition, the dual economic structure of urban and rural causes the endowment of household resources and the education level of farmers less than in the city, which affects the increase of rural residents' income and increases the income gap between urban and rural residents.

Thanks!

Reviewer #1comments5:

Conclusion reiterates the results, which is completely undesirable. The authors should summarize the results within a maximum of 3 sentences. Moreover, the policies are completely vague, and it seems that the authors already had the policies in mind before even starting the paper. The policies should be directly derived from the discussion of the results, and they should not go beyond the results.

Response: Thanks for your valuable comments and suggestions. We have re-summarized the results within 3 sentences in the conclusion policy suggestion. Moreover, we have deleted, supplemented, and revised the corresponding policy recommendations based on the research conclusion of this paper. The details are as follows:

Conclusion and policy suggestion

There is no relevant research on the relationship between financial development, technological innovation and urban-rural income gap in the existing literature. In the current environment of steady economic development and building a harmonious society, narrowing the urban-rural income gap is an important problem in the process of economic development in China. Therefore, this study investigates the FKC for urban-rural income gap in case of China for the period of 1985-2019. This study has intensified on the technological innovation-income gap link with the FKC. For this purpose, we apply the ARDL approach and Johansen method for cointegration. In addition, the long run coefficient estimates are conducted by DOLS, FMOLS and CCR estimators. We also apply the VECM Granger procedure to causality. Finally, we using OLS regression analysis to variables. Compared with existing research, this study has made improvements in the following two aspects: This paper will add the indicator of technological innovation. On the one hand, there is no literature on the relationship between the three in China. On the other hand, technological innovation is closely related to financial development and the income gap between urban and rural residents. Thus, the level of technological innovation cannot be abandoned in the study; It will verify the FKC theory and analyze the link between technological innovation and urban-rural income gap under the FKC hypothesis. 

 It is found that the long-run relationship exists among the variables under the structural breaks. The main finding obtained from the long-run coefficient estimates reveal that technological innovation increases urban-rural income gap. The findings confirm the validity of the FKC hypothesis for China’s economy in the long run. The causality analysis shows a bi-directional causality between financial development, technological innovation and urban-rural income gap in the long run. China's financial development bias is the main reason for this gap.

Based on the research conclusion of this paper, to further narrow the income gap between urban and rural areas, it is necessary to reduce the harmful effects of the technological innovation development process, and actively promote rural areas to build a new framework for economic development; Meanwhile, it is necessary to reverse the financial development bias and coordinate the balanced development of urban and rural finance. Therefore, the following countermeasures and suggestions are put forward:

First, the optimization and improvement of financial structure have a significant promotion effect on the main measurement indicators of technological innovation such as R&D investment and patent activities of large enterprises, but this may also be one of the main reasons for the intensification of urban-rural income gap in developing countries. Therefore, the government should take practical measures and introduce preferential policies to support technological innovation of small and medium-sized enterprises, create good financing conditions for small and medium-sized enterprises, and establish and develop technological service systems for small and medium-sized enterprises. In addition, local governments should insist on promoting industry to agriculture in technological innovation, cities supporting rural areas, and continuously increasing the intensity of financial investment in science and technology. On the one hand, while creating a good background to promote technological innovation, it should also introduce and implement relevant policies to promote technological innovation, create special innovation funds, actively subsidize and encourage scientific and technological innovation in large, medium, and small enterprises, research and development institutions, universities and other major scientific and technological innovation achievements. Vigorously promote industry-university-research cooperation between enterprises and universities, and help enterprises expand the market for technological innovation products and services. On the other hand, actively establish and improve the agricultural technology innovation coordination mechanism, improve the construction of agricultural technology innovation software and hardware, raise the level of agricultural technology, and lay a solid foundation for the construction of new rural industries.

Second, expand the scale of rural financial development and improve the efficiency of rural financial development. Because of the difficulty of loans in rural areas and the outflow of rural funds from rural areas through household savings and savings of township enterprises, when farmers have financing needs, they cannot get corresponding financing support because they cannot meet the credit threshold of financial institutions, and the efficiency of rural financial allocation is low. Therefore, the government should establish a sound agricultural financial system, optimize the share of agricultural financial services and financial resources, and try to improve the uneven distribution of financial resources between urban and rural areas and financial development to benefit more high-income people and less rural residents. Increase preferential policy support for rural economic development, improve the rural financial organization system, and promote agricultural modernization. In addition, the continuous expansion and upgrading of rural agricultural financial services will help attract more financial institutions and financial products to enter the countryside, and will also help rural areas reduce financing costs and thresholds, and reduce the loss of rural financial resources and talents. The improvement of rural financial operation efficiency can convert the absorbed rural savings into rural loans in real time, increase support for 'agriculture, rural areas and farmers' funds, increase the utilization rate of rural financial resources, and improve the development of rural financial development.

Third, establish and improve the system of rural financial security, guide and standardize the benign operation and development of rural areas. Under the background of the new era, with the rapid development and popularization of the Internet and e-commerce, the development potential of rural areas has gradually emerged. For example, at present, the Internet financial platform 'agriculture, rural areas and farmers' in China has continued to increase, the development of service industry has become increasingly diversified, and the business form has continued to evolve. In order to promote the benign operation and development of new rural finance, it is necessary to construct the institutional norms of rural financial market, a unified and effective financial supervision system and risk early warning system. 

Last, many macroeconomic factors are also closely related to urban-rural income gap. For example, high inflation, interest rates, and exchange rates in China's economic development have exacerbated income distribution. Therefore, government decision-makers should ensure economic and financial stability. 

This paper has some limitations. First, since we strengthen the relationship between technological innovation and urban-rural income gap in the context of FKC hypothesis, several explanatory variables such as globalization, human capital, and renewable energy are not included in our specifications. Second, many indicators of financial developments such as current liabilities and financial development index are not used for empirical analysis. At the same time, the study may inspire future researches. In this paper, there are some limitations in the relationship between patent licensing data analysis and urban-rural income gap. In the follow-up study, independent technological innovation indicators such as high-tech product export and R&D investment can be considered to further demonstrate the impact of technological innovation on urban-rural income gap and the differences in the results of different indicators. Therefore, future researches can investigate the impact of technological innovation on urban-rural income gap in detail and prefer comparative empirical results. 

Thanks once more!

Reviewer #2comments1: 

The results of causality analysis should be discussed in detailed with the findigs of other empirical studies.

Response: Thanks for your valuable comments and suggestions. We have discussed the results of causality analysis in detail with the findigs of other empirical studies. The details are as follows:

VECM Granger causality test

On the basis of the above test results, in the case of the optimal lag order of order 1, this section will continue to test the long-term and short-term causality of △lnGI、△lnTI and △lnFIR based on VECM. This method verifies the causality between variables in the composite system, which can avoid the disadvantage that the traditional Granger causality test cannot be applied to the cointegration test. The causality results presented in Table 7 suggest that financial development and income inequality cause each other. This conclusion is the same as that of Zhang Yingli and Yang Zhengyong [43]. He uses the VECM model to dynamically analyze the relationship between financial development, urbanization, and the urban-rural income gap. The empirical results show that financial development is the uni-directional Granger cause of the urban-rural income gap. The causal relationship results also show that there is a bidirectional causal linkage between technological innovation and urban-rural income gap at the significance levels of 1% and 5%. Ma Lei explored the impact of human capital structure and total factor productivity on urban-rural income gap from the perspective of innovation-driven development [29]. However, there is no research on the causal relationship between technological innovation and urban-rural income gap.

Table 7. VECM Granger causality test

Dependent variable Independent variable

Short-run Long-run

[p-value]

 F-statistic

[p-value] 

 △lnGI △lnTI △lnFIR ECTt-1

△lnGI ¬- 13.824***

[0.0008] 9.4808**

[0.0044] -0.0003**

[0.035]

△lnTI 7.4338**

[0.0107] - 7.5969***

[0.0098] -0.0015**

[0.006]

△lnFIR 5.0151**

[0.0327] 4.2866**

[0.0471] - -0.0010***

[0.000]

Note: *** and **denote significance at 1% and 5% levels, respectively.

Overall, short-term fluctuations and long-term equilibrium characterize the relationship between financial development, technological innovation, and the urban-rural income gap. Financial development and technological innovation will have an impact on the urban-rural income gap, which is the result of China's financial development bias, and it is also an inevitable phenomenon that the process of technological innovation and development has an impact on the income distribution pattern of different groups. 

The bias in financial development has had an impact on the expansion of the urban-rural income gap in China to a certain extent. On the one hand, the profit orientation of capital and China's financial policies focus on supporting urbanization, resulting in a part of rural funds entering the urban financial market, accelerating the economic development of cities and the increase of urban residents' income, but not affecting rural construction, hindering the development and growth of the rural economy. On the other hand, when rural finance lags behind the development of urban finance, due to the imperfect rural financial market and mechanism, the low level of rural investment and consumption, coupled with the relatively weak government support for rural finance to agricultural development, the agricultural support function of financial institutions has been seriously degraded, resulting in the phenomenon that financial institutions are 'retreating weaker and weaker, weaker and retreating' in rural areas, leading to the development dilemma of small-scale and low efficiency of rural financial markets, these long-term constraints lead to narrowing the income gap between urban and rural residents has become quite difficult.

Due to the 'urban-rural dual' structure of the national industrial policy, there are differences in scientific and technological innovation ability and innovation efficiency between urban and rural areas.

Agricultural scientific and technological innovation does not match human capital, the accumulation rate of agricultural high-quality human capital lags behind the needs of technological innovation, and agricultural technological innovation has a weak impact on the urban-rural income gap; The urban industrial sector and science and technology service sector have an enormous investment in innovation resources, high scientific and technological innovation ability, high innovation efficiency, large profit space, and rapid production efficiency improvement. In addition, the dual economic structure of urban and rural causes the endowment of household resources and the education level of farmers less than in the city, which affects the increase of rural residents' income and increases the income gap between urban and rural residents.

Thanks!

Reviewer #2comments2: 

To improve the empirical litarature, the following references should be integrated into the literature.

Response: Thanks for your valuable comments and suggestions. We have integrated the following references into the literature. The details are as follows:

References：

1. Zhenmei Hou, Maozai Tian, Zhihao Wang, Yan Dou. Study on the impact of technological innovation on the income gap between urban and rural residents—Based on the empirical analysis of Western Ethnic agglomeration. Practice and understanding of mathematics. 2020; 50(02): 53–64.

2. Pan X, Uddin M K, Ai B, Pan X, Saima U. Influential factors of carbon emissions intensity in OECD countries: evidence from symbolic regression. Journal of Cleaner Production. 2019; 220,1194–1201. https://doi.org/10.1016/j.jclepro.2019.02.195.

3. Huiping W, Meixia W. Effects of technological innovation on energy efficiency in China: evidence from dynamic panel of 284 cities. Science of The Total Environment. 2020; 709,136172. https://doi.org/10.1016/j.scitotenv.2019.136172.

4. Yii K-J, Geetha C. The nexus between technology innovation and CO2 emissions in Malaysia: evidence from granger causality test. Energy Procedia. 2017; 105,3118–3124. https://doi.org/10.1016/j.egypro.2017.03.654.

5. Peng Zeng, Gongliang Wu. Technological Progress, Industrial Agglomeration, Urban Scale and Urban-Rural Income Gap.Journal of Chongqing University (Social Science Edition). 2015; 21(06):18–34. 

6. Zheng Zhao, Liangliang Zhang, Zhi Chen. Technological innovation, urbanization and urban rural income gap: An Empirical Analysis Based on urban panel data. China Science and Technology Forum.2018; (10):138–145.doi: 10.13580/j.cnki.fstc.2018.10.016.

7. Kuznets S. Economic growth and income inequality. The Economic Review.1995; 45(1),1–28.

8. Greenwood J, Jovanovich B. Financial development, growth, and the distribution of income. Journal of Political Economy. 1990; 98(5), 1076–1107.

9. Townsend R M, Ueda K. Financial Deepening, Inequality, and Growth: A Model—Based Quantitative Evaluation. The Review of Economic Studies.2006; 73(1):251–293. https://doi.org/10.1111/j.1467-937X.2006.00376.x.

10. Clarke, George RG, Heng-fu Z, Lixin Colin X. Finance and income inequality: test of alternative theories. Vol. 2984. World Bank Publications. 2003.

11. Pradhan R P. The nexus between finance, growth and poverty in India: The cointegration and causality approach. Asian Social Science. 2010; 6(9):114.

12. Law S H, Tan H B, Azman-Saini W N W. Financial development and income inequality at different levels of institutional quality. Emerging Markets Finance and Trade. 2014; 50(sup1),21–33. https://doi.org/10.1016/j.jclepro.2019.02.195.

13. Sehrawat M, Giri, A K. Financial development and income inequality in India: an application of ARDL approach. International Journal of Social Economics.2015; 42(1):64–81. https://doi.org/10.1108/IJSE-09-2013-0208.

14. Haishu Qiao, Li Chen. Re test of the inverted U-shaped relationship between financial development and urban-rural income gap: An Empirical Analysis Based on China's County cross-sectional data. China's rural economy.2009; (07):68–76+85.

15. Zongyi Hu, Yiwen Liu. Research on Kuznets Effect of unbalanced financial development and urban-rural income Gap Empirical Analysis Based on cross-sectional data of counties in China. Statistical research.2010; 27(05):25–31.doi:10.3969/j.issn.1002-4565.2010.05.004. 

16. Nan Yang, Chuoxin Ma. The dynamic inverted U evolution of the impact of China's financial development on the urban-rural income gap and the prediction of its decline point. Financial research. 2014; (11):175–190.

17. Zhiqiang Ye, Xiding Chen, Shunming Zhang. Can financial development reduce the income gap between urban and rural areas-Evidence from China. Financial research.2011; (02):42–56.

18. Fei Jia. Test on the unbalanced effect of financial development on urban rural income gap. Statistics and decision making. 2015; (24): 92–95.doi:10.13546/j.cnki.tjyjc.2015.24.026.

19. Yongqiang Sun, Yulin Wan. Financial Development, Opening Up and Income Gap between Urban and Rural Residents: An Empirical Analysis Based on Interprovincial Panel Data from 1978 to 2008. Financial Research. 2011; (01):28–39.

20. Youcai Yang. Financial Development and Economic Growth: An Analysis Based on the Threshold Variables of China's Financial Development. Journal of Financial Research. 2014; (02):59–71.

21. Edward E, Leamer. What’s the use of factor contents? Journal of International Economics.2000; 50(1). https://doi.org/10.1016/S0022-19-96(99)00004-5.

22. Antonio C. Agglomeration effects in European. European Economic Review.2002; 46:213-227. https://doi.org/10.1016/S0014-2921(00)00099-4.

23. Cirillo V, Corsi M, D’Ippoliti C. European households’incomes since the crisis. Investigaci6n Economica.2017; 76(301):57–85. https://doi.org/10.1016/j.inveco.2017.12.002

24. Nathan M, Lee N. Cultural Diversity, Innovation, and Entrepreneurship: Firm-level Evidence from London. Economic Geography.2013; 89(4):367–394. 

25. Aghion P, Akcigit U, Bergeaud A, Blundell R, Hémous D. Innovation and top income inequality. The Review of Economic Studies.2019; 86(1),1–45. https://doi.org/10.1093/restud/rdy027.

26. Yong Chen, Zhe Bai. Skill-biased technological progress, labor agglomeration effect and widening of regional wage gap. China Industrial Economy. 2018; (09):79–97. doi:10.19581/j.cnki.ciejournal.2018.09.015.

27. Peng Zeng, Gongliang Wu, Xiaojun Zang. Research on the Relationship between Technological Progress, Urbanization and Urban-Rural Income Gap: Based on empirical analysis of Urban Agglomerations in China. East China Economic Management. 2016; 30(02):64–70.

28. Zhiqiang, Dong, Xiao Cai, Linhui Wang. Skill Premium: an explanation based on the direction of technological progress. Chinese Social Sciences. 2014; (10):22–40+205-206.

29. Lei Ma. Human capital structure, technological progress and urban rural income gap: an analysis based on the panel data of 30 provinces in China from 2002 to 2013. East China economic management. 2016; 30(02):56–63.doi:10.3969/j.issn.1007-5097.2016.02.010.

30. Dickey D A, Fuller, W A. Likelihood ratio statistics for autoregressive time series with a unit root. Econometrica.1981; 49(4),1057–1072. https://doi.org/10.2307/1912517.

31. Phillips P C B, Perron P. Testing for a unit root in time series regression. Biometrika.1988; 75(2),335–346. https://doi.org/10.1093/biomet/75.2.335

32. Elliott G, Rothenberg, T J, Stock J H. Efficient tests for an autoregressive unit root. Econo-metrica.1996; 64(4),813–836. https://doi.org/10.2307/2171846.

33. Pesaran M H, Shin Y, Smith R. Bounds testing approaches to the analysis of level relationship. Journal of Applied Econometrics, 2001; (3):289-326. https://doi.org/10.1002/jae.616.

34. Stock J H, Watson M W. A simple estimator of cointegrating vectors in higher order integrated systems. Econometrica.1993; 61(4),783–820. https://doi.org/10.2307/2951763.

35. Phillips P C B, Hansen B E. Statistical inference in instrumental variables regression with I(1) processes. The Review of Economic Studies. 1990; 57(1),99–125. https://doi.org/10.2307/2297545.

36. Park J Y. Canonical cointegrating regressions. Econometrica.1992; 60(1),119–143. http://www.jstor.org/stable/2951679.

37. Engle R F, Granger C W J. Co-integration and error correction: representation, estimation, and testing. Econometrica.1987; 55(2), 251–276. https://doi.org/10.2307/1913236.

38. Jing Wang. An analysis of the relationship between Taiwan's economic openness and income gap—An Empirical Study Based on ardl-ecm model. Industrial economic circles. 2016; 03(02):164–172.

39. Bildirici M E. The effects of militarization on biofuel consumption and CO2 emission. Journal of Cleaner Production.2017; 152,420–428. https://doi.org/10.1016/j.jclepro.2017.03.103.

40. Fan Ye, Dongtao Zou, Xiheng Yuan. The impact of economic financialization on the income gap between urban and rural areas in China: an analysis based on provincial panel data from 1978 to 2013. Contemporary economic science. 2015; 37 (06):61–67+124.

41. Aghion P, Akcigit U, Bergeaud A, Blundell R, Hemous D. Innovation and top income inequality. Review of Economic Studies.2019; 86(1),1–45.https://doi.org/10.1093/restud/rdy027.

42. Mnif S. Bilateral relationship between technological changes and income inequality in developing countries. Atlantic Review of Economics.2016; 1(1),4.

43. Yingli Zhang, Zhengyong Yang. Mechanism and dynamic analysis of financial development and urbanization on urban and rural income gap. Statistics and Decision Making. 2018; 34(05):84-88. doi:10.13546/j.cnki.tjyjc.2018.05.020.

44. Antonelli C, Gehringer A. Technological change, rent and income inequalities: a Schumpeterian approach. Technological Forecasting and Social Change.2017; 115, 85–98. https://doi.org/10.1016/j.techfore.2016.09.023.

45. Zhihuan Chen, Hongge Zhu, Bo Cao. Financial Development, Economic Growth, and Energy Consumption Analysis is based on the ARDL-ECM model. Technical Economy and Management Research. 2020; (05):97–104.

46. Tao Feng, Maoguang Wu, Meisha Zhang. Financial development, industrial structure and urban-rural income gap: an analysis based on the perspective of "from real to virtual" finance. Exploration of economic problems. 2020; (10):170–181.

47. Shulan Fei, Jiqiang Guo. The impact of the statistical income attribution of migrant workers on the income gap between urban and rural areas. Statistical study. 2014; 31(06):17–24.

48. Jinpei Liu, Xiaoxia Song, Huayou Chen, Guanzhen Wang, Zhen Wang. Study on long-term equilibrium and causal dynamics of influencing factors affecting per capita carbon emissions in China by based on the empirical analysis of structural mutation ARDL-VECM model. Operations Preparation and Management. 2019; 28(09):57–65.

49. Ping Li, Tinghua Liu. Research on the Relations between Technology Innovation and Income Inquality—Based on China Data. Industrial Technical economy. 2009; 1:41–46.

50. Jingwen Li, Zhengdong Yang. The impact of technological innovation on industrial structure and income distribution. Journal of Social Science of Jilin University. 2013; (6):5–11.doi:10.15939/j.jujsse.2013.06.014.

51. Madhu S H W, Giri A K. Financial development and income inequality in India: an application of ARDL approach. Social Indicators Research.2011; 41(2):1-26. https://doi.org/10.1108/IJSE-09-2013-0208.

52. Huaizhong Mu, Peng Wu. Urbanization, industrial structure optimization and urban and rural income gap. Economist. 2016; (05):37–44.doi: 10.16158/j.cnki.51-1312/f.2016.05.006.

53. Pece A M, Simona O E O, Salisteanu F. Innovation and economic growth: an empirical analysis for CEE countries. Procedia Economics and Finance.2015; 26,461–467. https://doi.org/10.1016/S2212-5671(15)008746.

54. Permana M Y, Lantu D C, Suharto, Y. The effect of innovation and technological special-ization on income inequality. Problems and Perspectives in Management.2018; 16(4):51–63. 

55. Risso W A, Carrera E J S. On the impact of innovation and inequality in economic growth. Economics of Innovation and New Technology. 2019; 28(1):64–81. https://doi.org/10.1080/10438599.2018.1429534.

56. Sohag K, Begum R A. Abdullah, S.M.S. et al. Dynamics of energy use, technological innovation, economic growth and trade openness in Malaysia. Energy.2015; 90 (Part2):1497–1507. https://doi.org/10.1016/j.energy.2015.06.101.

57. Shahbaz M, Raghutla C, Song M, Zameer, H, Jiao Z. Public-private partnerships investment in energy as new determinant of CO2 emissions: the role of technological innovations in China. Energy Economics.2020; 86,104664. https://doi.org/10.1016/j.104664.

58. Thornton J, Tommaso C D. The long-run relationship between finance and income inequal-ity: evidence from panel data. Finance Research Letters. 2020; 32,101180. https://doi.org/10.1016/j.frl.2019.04.036.

59. Xiang Wang, Wang, Qingfang. Urbanization, service growth and urban and rural income gap-based on cointegration approach and structural equations. Modern Finance and Economics (Journal of Tianjin University of Finance and Economics). 2013; 33(06):45-56.doi:10.19559/j.cnki.12-1387.2013.06.005.

60. Zhang, Yantao. Energy Consumption and Economic Growth Restudy—based on FMOLS, DOLS, CCR Research. Journal of North China University of Electric Power (Social Sciences). 2012; (01):5–9.

61. Cetin M, Demir H, Saygin S. Financial development, technological innovation and income inequality: Time series evidence from Turkey. Social Indicators Research.2011; 156(1),47–69. https://doi.org/10.1007/s11205-021-02641-7.

62. Ozturk S, Cetin M, Demir H. Income inequality and CO2 emissions: nonlinear evidence from Turkey. En-vironment, Development and Sustainability.2021; 1–18. https://doi.org/10.1007/s10668-021-01922-y.

63. Altunbaş Y, Thornton J. The impact of financial development on income inequality: A quantile regression approach. Economics Letters.2019; 175,51–56. https://doi.org/10.1016/j.econlet.2018.12.030.

Thanks once more!

Editors #comments1: 

Response: Thanks for your valuable comments and suggestions. We have modified the format of the manuscript and ensured that your manuscript meets PLOS ONE's style requirements.

Thanks!

Editors #comments2: 

Thank you for stating the following in your Competing Interests section: 

The authors declare that they have no known competing financial interests or personal relationships that could have appeared to influence the work reported in this paper. Please complete your Competing Interests on the online submission form to state any Competing Interests. If you have no competing interests, please state "The authors have declared that no competing interests exist.", as detailed online in our guide for authors at http://journals.plos.org/plosone/s/submit-now

Response: Thanks for your valuable comments and suggestions. We have added this information to our cover letter.

Thanks!

Editor #comments3:

We note you have included a table to which you do not refer in the text of your manuscript. Please ensure that you refer to Table 2 in your text; if accepted, production will need this reference to link the reader to the Table.

Response: Thanks for your valuable comments and suggestions. We have made modifications in the text of your manuscript. The details are as follows:

Empirical results and discussion

Unit root test

To avoid the phenomenon of pseudo-regression, the variables of the time series must be stable before constructing the dynamic econometric model. This study uses ADF and PP as two test methods to test the unit root stationery of the time series of the variables one by one. From the test results (see Table 2), at the significance level of 1%, the t statistical values of the ADF and PP tests of lnTI are−16.04485 and−13.7094, which are less than the critical values. Therefore, the null hypothesis is rejected. There is no unit root in lnTI, which is the zero-order unitary sequence I(0), the original sequence is stable. However, the first-order difference series of lnGI, lnFIR and lnFIR2 are stable at a 1% significant level, belonging to the first-order mono integral sequence I(1). In order to analyze the cointegration relationship of the same order difference sequence, we carried the cointegration test out below.

Table 2. Unit root test results

Varizbles Form ADF

t-atatistic PP

Adj.t-atatistic Result

lnGI (C,T,1) -1.661718 -2.77091 -

lnTI (C,T,1) -16.04485*** -13.7094*** I(0)

lnFIR (C,T,1) -2.745414 -2.88569 -

lnFIR2 (C,T,1) -2.591233 -2.72779 -

△lnGI (C,T,1) -4.474620*** -4.50768*** I(1)

△lnFIR (C,T,1) -5.519429*** -8.02233*** I(1)

△lnFIR2 (C,T,1) -5.56658*** -8.01153*** I(1)

Note: △ represents the first-order difference of variables;（C,T,K）

represents the intercept term, trend term and lag order of ADF resp-

ectively; ***represents rejection of the null hypothesis at the1% sig-

nificance level.

Thanks once more!

---

## [Decision Letter · Decision Letter 1]

11 Aug 2022

PONE-D-22-09786R1Financial development, technological innovation and urban-rural income gap: time series evidence from ChinaPLOS ONE

Dear Dr. Wu,

Thank you for submitting your manuscript to PLOS ONE. After careful consideration, we feel that it has merit but does not fully meet PLOS ONE’s publication criteria as it currently stands. Therefore, we invite you to submit a revised version of the manuscript that addresses the points raised during the review process.

We look forward to receiving your revised manuscript.

Kind regards,

Ming Zhang, Ph.D.

Academic Editor

PLOS ONE

Journal Requirements:

Reviewers' comments:

Reviewer's Responses to Questions

**Comments to the Author**

1. If the authors have adequately addressed your comments raised in a previous round of review and you feel that this manuscript is now acceptable for publication, you may indicate that here to bypass the “Comments to the Author” section, enter your conflict of interest statement in the “Confidential to Editor” section, and submit your "Accept" recommendation.

Reviewer #1: (No Response)

Reviewer #2: All comments have been addressed

2. Is the manuscript technically sound, and do the data support the conclusions?

Reviewer #1: No

Reviewer #2: Yes

3. Has the statistical analysis been performed appropriately and rigorously? 

Reviewer #1: No

Reviewer #2: Yes

4. Have the authors made all data underlying the findings in their manuscript fully available?

Reviewer #1: No

Reviewer #2: Yes

5. Is the manuscript presented in an intelligible fashion and written in standard English?

Reviewer #1: No

Reviewer #2: Yes

6. Review Comments to the Author

Reviewer #1: The authors were not able to fully comprehend the comments.

1. Why the authors think that FKC estimation will be a suitable theoretical framework for addressing such a research question? The objective of the study as stated by the authors is "to focus on verifying the validity of the FKC hypothesis and the dynamic relationship between technological innovation and the urban-rural income gap in the FKC hypothesis". It goes back to the first comment I have given. Authors are basically carrying out a FKC estimation study, without having any solid policy relevance.

2. The research gaps are vague. The findings of FKC estimation studies are already inconsistent, and the present study conveniently adds to that inconsistency. Moreover, merely adding a variable in the framework cannot be considered as a valid research gap.

3. The reduced form model is susceptible to omitted variable bias and multicollinearity issues. These issues are not addressed.

4. The discussion is superficial, jus like the previous version.

5. The study should be summarized in just THREE sentences at the beginning of the conclusion.

6. The policy recommendations are vague. One does not need to carry out an entire study to arrive at these policy recommendations.

Reviewer #2: I have demanded two topics. All the recommendations have been performed. Thank you very much for answers of author.

7. PLOS authors have the option to publish the peer review history of their article (what does this mean?). If published, this will include your full peer review and any attached files.

Reviewer #1: No

Reviewer #2: No

---

## [Author Response · Author response to Decision Letter 1]

21 Sep 2022

Thank you for your letter and for the reviewers' and editors' comments concerning our manuscript. These comments are very helpful for revising and improving our paper. We have studied comments carefully and have made corrections which we hope meet with approval.

Reviewer #1comments1: 

Why the authors think that FKC estimation will be a suitable theoretical framework for addressing such a research question? The objective of the study as stated by the authors is "to focus on verifying the validity of the FKC hypothesis and the dynamic relationship between technological innovation and the urban-rural income gap in the FKC hypothesis". It goes back to the first comment I have given. Authors are basically carrying out a FKC estimation study, without having any solid policy relevance.

Response: Thanks for your valuable comments and suggestions. We have revised and supplemented the introduction, including the necessity of this study, the policy issues to be solved by the study, and the reasons for the selection of research samples. To stimulate research. At the same time, On the basis of the estimation study of FKC hypothesis, some relevant and reliable policies are supplemented. The details are as follows:

Introduction

Income distribution has always been a popular issue in entire society. A reasonable income distribution system and income gap are not only the common aspiration of the people but also an important embodiment of social justice. China's urban-rural dual structure problem results from the country's policy choice in the process of reform and opening up. The country's city priority development strategy and the opening-up pilot program of coastal cities and regions have attracted an influx of talents and funds from inland areas, which has led to further disparities in income distribution between urban and rural areas and regions. Especially in recent years, with the widening gap between urban and rural residents' income growth rate, the income gap between urban and rural areas in China continues to increase, which has become an obstacle to the long-term healthy and stable development of China's economy. The data from China Statistical Bureau show that the income ratio of urban and rural residents in China has decreased from 3.30: 1 in 2007 to 2.64: 1 in 2019, and the overall trend is declining. However, compared with the world level, the income gap between urban and rural areas in China is hovering at a high level and shows a fluctuating trend. During 2002-2019, it expanded at a high level before 2009, and then fell within a narrow range, showing an inverted U-shaped trend. The Fifth Plenary Session of the 19th Central Committee of the Communist Party of China proposed the long-term goals for 2035, including 'more obvious and substantial progress in common prosperity for all people' and 'per capita GDP reaches the level of moderately developed countries, and the middle-income group has significantly expanded'. At present, the government has introduced a variety of agricultural financial policies to promote rural economic development, and gradually relaxed the household registration management system to strengthen labor mobility. However, under the current urban-rural dual structure system, the income distribution of residents in China is still very serious, and the income gap between urban and rural residents is expanding, which will inevitably affect social harmony and stability. Therefore, in realizing the path of establishing and improving the institutional mechanism of urban-rural integration development and balancing urban-rural integration development put forward in the 19th National Congress of the Communist Party of China, solving the urban-rural income gap has become the key point the sustainable economic development at this stage and in the future for a long time.

Following Kuznets [1], Greenwood and Jovanovich [2] first reveal an inverted U-shaped relationship between financial development and income inequality. Namely, with the development of finance, the income gap shows an inverted U-shaped feature of early expansion and then reduction, which is the financial Kuznets curve (FKC) hypothesis. In the early 1990s, the income gap between countries in the world was highlighted, and the mechanism of financial development on income distribution attracted wide attention in academia. At present, the research in this area has become a research frontier of financial development theory. Looking at the actual situation of our country, since the reform and opening up, the financial market has developed rapidly, the reform of the financial industry has achieved positive development, and the innovation of the industry has gradually deepened. In 2018, the scale of social financing reached 200.75 trillion yuan in the entire year, and the scale of the interbank bond market leaped to the second in the world. However, with the rapid development of financial markets, accompanied by the widening income gap between urban and rural residents, the problem of unbalanced and inadequate development still exists. Furthermore, technological innovation is considered to be the major factor affecting macroeconomic variables. Hou Zhenmei, Tian Mao, et al. [3] pointed out that the process of technological innovation is an economic growth process that takes technology as the mainline and causes social change and technological innovation is a key driving force for economic growth, and its development process will inevitably affect the change of income distribution pattern of different groups. Pan et al. [4], Wang and Wang [5] dwell on the influence of technological innovation on energy efficiency. Technological innovation can influence environmental degradation. Recently, the effect of innovation on the income gap is an important field of research [6]. 

The China's economy is an essential case. In 2016, China became the first middle-income economy to enter the top 25 of the global innovation index. China's scientific and technological innovation capability was enhanced and the main scientific and technological innovation indicators were steadily improved in 2018. The state intellectual property office of the data shows the proportion of research and experimental development expenditure of the entire society in GDP in 2018 was 2.15%. The total number of R&D personnel reached 4.18 million person-years, ranking first in the world; The number of international scientific papers and citations ranked second in the world; The number of patent applications and authorization of invention ranks first in the world; The contribution rate of scientific and technological progress is expected to exceed 58.5%, and the national comprehensive innovation ability ranks 17th in the world. The level of scientific and technological innovation in China has been continuously improved, and fruitful achievements have been made in many fields such as innovation input, innovation output, and innovation efficiency. These developments show technological innovations advance rapidly in China. In addition, income distribution has been a vital matter in China for a long time. The report of the 19th National Congress of the Communist Party of China pointed out that my country's 'urban and rural regional development and income distribution gap is still large', which must be solved. In this context, to incorporate technological innovation into the important factors that affect the income gap between urban and rural areas, and to explore ways and channels to ease and eliminate the excessive income gap in residential areas in China, which is of great significance to reduce income inequality and realize social equity at this stage. 

Considering all the above explanations, the present study creates the following questions: First, is the FKC hypothesis valid for China? Second, how technological innovation influences income gap in China's economy? Third, is there a causal linkage between technological innovation and urban-rural income gap? In addition, there are no studies on the causal linkage between them. In this regard, this is the first study on China's economy. To answer these questions above, we attempt to make an empirical study on the link between technological innovation and income gap in China from 1985 to 2019 under the context of FKC hypothesis. We intensify on the long-term and causal relations between the variables. In the study of financial development and the income gap between urban and rural residents, this paper adds the index of technological innovation. By citing these three variables, hope to be more comprehensive on the causes of China 's urban-rural income gap and how to solve this problem are discussed, and thus draw more accurate results and set forward more constructive suggestions. The widening income gap between urban and rural areas in China has attracted wide attention from the government and academia, but from the domestic existing literature, it focuses on the role of law decision-making factors. There is no analysis on the impact of technological innovation on the income gap under the background of FKC hypothesis. In addition, the utility research on the long- and short-term causality test between technological innovation and income gap is indeed more inadequate. Thus, the study intends to focus on verifying the validity of the FKC hypothesis and the dynamic relationship between technological innovation and the urban-rural income gap in the FKC hypothesis. It provides the theoretical basis and policy suggestions for further deepening rural financial reform and technological innovation and giving full play to the role of financial services and technological innovation in the allocation of rural resources. This newly explains the causes of the urban-rural income gap, which is not only related to the overall promotion of the coordinated development of urban and rural areas but also the determination of the focus and specific direction of future financial and technological innovation development in various regions. Therefore, it is an important subject worthy of in-depth study and testing.

Thanks! 

Reviewer #1comments2: 

The research gaps are vague. The findings of FKC estimation studies are already inconsistent, and the present study conveniently adds to that inconsistency. Moreover, merely adding a variable in the framework cannot be considered as a valid research gap.

Response: Thanks for your valuable comments and suggestions. According to the research framework, core variables and research significance of this study, my views are as follows:

1）In the study of financial development and urban-rural income gap, there are three dominant views: i) Inverted U-shaped action view. Greenwood and Jovanovich first reveal an inverted U-shaped relationship between financial development and income inequality. Namely, with the development of finance, the income gap shows an inverted U-shaped feature of early expansion and then reduction, which is the financial Kuznets curve (FKC) hypothesis. ii) Expand the role of view. Based on the extension of Greenwood and Jovanovich model, Calor and Zeira, Banerjee and Newman put forward the view that capital market imperfections will widen the income gap. iii) Inhibition view. Chakraborty and Ray proposed that the bank dominated financial system is conducive to narrowing the income gap.

2）Innovation is the long-term driving force of economic growth and has a profound impact on economic development. Technological innovation becomes the core strategy of regional economic transformation and development. Exploring the impact of technological innovation on the urban-rural income gap is of some significance. Technological innovation is considered being the major factor affecting macroeconomic variables. However, the existing studies do not consider technological innovation as a factor determining income inequality in term of the FKC. Thus, due to deficiencies in the literature, the present study aims at examining the relationship between technological innovation and urban-rural income gap in the presence of FKC hypothesis for China economy. Therefore, only the core variable of technological innovation is added as the key analysis object of the study. 

Thanks!

Reviewer #1comments3:

The reduced form model is susceptible to omitted variable bias and multicollinearity issues. These issues are not addressed.

Response: Thanks for your valuable comments and suggestions. According to the model selection, omitted variable bias and multicollinearity related problems in the manuscript, my views are as follows:

The empirical strategy of this study mainly consists of four different stages. We first apply the ADF and PP unit root tests presented by Dickey and Fuller, Phillips and Perron and Elliott et al. for stationary analysis. 

Second, the cointegration methods（ARDL）proposed by Pesaran et al. are used to determine the cointegration between the series. From the research of Wang Jing, we can see that the ARDL test is superior to other classical cointegration tests the approach does not require the variable data to be a single integer sequence of the same order. After determining the optimal lag order, ARDL can analyze the long-term relationship of variables regardless of whether the variables are I(0), I(1) or mixed sequences.

Third, the long-run coefficients are estimated through the DOLS, FMOLS, and CCR estimation methods developed by Stock and Watson, Phillips and Hansen, and Park, respectively. According to Zhang Yantao's research, the DOLS technique is an asymptotically effective estimator that eliminates autocorrelation, and simultaneity problems in the cointegration equation. The FMOLS has the advantage of correcting autoregression and endogeneity problems. It is well known that omitted variables are one of the causes of endogeneity problems, so the model in this study will not be affected by omitted variable bias.

In the last stage, the causal relations between the series are analyzed by the VECM Granger causality method presented by Engle and Granger. Engle and Granger（1987）add the error correction term (ECT) in the classical VAR model as an additional variable. It has the advantage of being able to analyze the long-term and short-term causality between variables.

According to the relevant books on econometrics and website data, the application of the VECM model does not need to consider the problem of multicollinearity. The reasons are as follows：

i) The problem of multicollinearity itself affects the significance of a single factor, not the significance of the whole model. The purpose of the VECM model is not to explore the influence of individual factors on endogenous latent variables, but to analyze their comprehensive influence on endogenous latent variables as a whole, which is the key to the problem.

ii) The original intention of the VAR model is to construct the model by taking each endogenous variable as the lag value of all endogenous variables in the system, regardless of the economic theory itself.

Thanks once more!

Reviewer #1comments4:

The discussion is superficial, just like the previous version.

Response: Thanks for your valuable comments and suggestions. We have added economic intuitions and policy discussion behind the results. The details are as follows:

VECM Granger causality test

On the basis of the above test results, in the case of the optimal lag order of order 1, this section will continue to test the long-term and short-term causality of △lnGI、△lnTI and △lnFIR based on VECM. This method verifies the causality between variables in the composite system, which can avoid the disadvantage that the traditional Granger causality test can not be applied to the cointegration test. The causality results presented in Table7 suggest that financial development and income inequality cause each other. This conclusion is the same as that of Zhang Yingli and Yang Zhengyong [41]. He uses the VECM model to dynamically analyze the relationship between financial development, urbanization, and the urban-rural income gap. The empirical results show that financial development is the uni-directional Granger cause of the urban-rural income gap. The causal relationship results also show that there is a bidirectional causal linkage between technological innovation and urban-rural income gap at the significance levels of 1% and 5%. Ma Lei explored the impact of human capital structure and total factor productivity on urban-rural income gap from the perspective of innovation-driven development [27]. However, there is no research on the causal relationship between technological innovation and urban-rural income gap.

Table 7. VECM Granger causality test

Dependent variable Independent variable

Short-run Long-run

[p-value]

 F-statistic

[p-value] 

 △lnGI △lnTI △lnFIR ECTt-1

△lnGI ¬- 13.824***

[0.0008] 9.4808**

[0.0044] -0.0003**

[0.035]

△lnTI 7.4338**

[0.0107] - 7.5969***

[0.0098] -0.0015**

[0.006]

△lnFIR 5.0151**

[0.0327] 4.2866**

[0.0471] - -0.0010***

[0.000]

Note: *** and **denote significance at 1% and 5% levels, respectively.

Overall, short-term fluctuations and long-term equilibrium characterize the relationship between financial development, technological innovation, and the urban-rural income gap. Financial development and technological innovation will have an impact on the urban-rural income gap, which is the result of China's financial development bias, and it is also an inevitable phenomenon that the process of technological innovation and development has an impact on the income distribution pattern of different groups. 

The bias in financial development has had an impact on the expansion of the urban-rural income gap in China to a certain extent. On the one hand, the profit orientation of capital and China's financial policies focus on supporting urbanization, resulting in a part of rural funds entering the urban financial market, accelerating the economic development of cities and the increase of urban residents' income, but not affecting rural construction, hindering the development and growth of the rural economy. On the other hand, in the case of rural finance lagging behind urban financial development, due to the imperfect rural financial market and mechanism, the level of rural investment and consumption is low, which is not conducive to the sustainable development of the cause of agriculture, rural areas and farmers and reduces the income of rural residents. Coupled with the government's rural financial support for agricultural development being relatively weak, the serious degradation of financial institutions to support agriculture, financial institutions in rural areas 'retreating weaker and weaker, weaker and retreating' phenomenon, leading to the rural financial market into a small scale, low-efficiency development dilemma. These long-term constraints have made it very difficult to narrow the income gap between urban and rural residents and gradually separated from poverty alleviation and rural revitalization.

The different effects of financial development on state-owned enterprises and private enterprises are also closely related to the urban-rural income gap. Since the reform and opening up, China has adopted 'gradual reform', and one of the important strategies is the dual track reform strategy, that is, plan and market, state-owned enterprises and private enterprises coexist. In the early days of reform and opening up, the government took supporting the development of state-owned enterprises as its primary goal, and also actively provided policy support for the development of small and medium-sized enterprises and private enterprises. However, with the deepening of the reform, the problem of low operating efficiency of state-owned enterprises has gradually emerged. To support the continued operation of state-owned enterprises, government departments have provided disguised subsidies to state-owned enterprises in various ways. The bias of financial policies toward state-owned enterprises will inevitably lead to more difficult living spaces for small and micro enterprises or private enterprises. This shows that the services currently provided by China's financial industry cannot fully meet the financing needs of private enterprises and small and medium-sized enterprises, and the financial system still has obvious shortcomings. Although the dual-track reform is intended to support the common development of state-owned enterprises and private enterprises, the actual implementation of the financial policy still prefers state-owned enterprises, and SMEs can not be treated equally. Small and micro enterprises and private enterprises are difficult to promote their development by financial means, which is not conducive to alleviating the income gap.

Due to the 'urban-rural dual' structure of the national industrial policy, there are differences in scientific and technological innovation ability and innovation efficiency between urban and rural areas.

Agricultural scientific and technological innovation does not match human capital, the accumulation rate of agricultural high-quality human capital lags behind the needs of technological innovation, and agricultural technological innovation has a weak impact on the urban-rural income gap; The urban industrial sector and science and technology service sector have an enormous investment in innovation resources, high scientific and technological innovation ability, high innovation efficiency, large profit space, and rapid production efficiency improvement. In addition, the dual economic structure of urban and rural causes the endowment of household resources and the education level of farmers less than in the city, which affects the increase of rural residents' income and increases the income gap between urban and rural residents.

Thanks!

Reviewer #1comments5:

The study should be summarized in just THREE sentences at the beginning of the conclusion.

Response: Thank you very much for your reminder. We have summarized the conclusion in three sentences at the beginning of this manuscript. Please see the revised version of manuscript for details. 

Thanks once more!

Reviewer #1comments6:

The policy recommendations are vague. One does not need to carry out an entire study to arrive at these policy recommendations.

Response: Thanks for your valuable comments and suggestions. We have deleted, supplemented, and revised the corresponding policy recommendations based on the research conclusion of this manuscript. The details are as follows:

Conclusion and policy suggestion

There is no relevant research on the relationship between financial development, technological innovation and urban-rural income gap in the existing literature. In the current environment of steady economic development and building a harmonious society, narrowing the urban-rural income gap is an important problem in the process of economic development in China. Therefore, this study investigates the FKC for urban-rural income gap in case of China for the period of 1985-2019. This study has intensified on the technological innovation-income gap link with the FKC. For this purpose, we apply the ARDL approach and Johansen method for cointegration. In addition, the long run coefficient estimates are conducted by DOLS, FMOLS and CCR estimators. We also apply the VECM Granger procedure to causality. Finally, we using OLS regression analysis to variables. Compared with existing research, this study has made improvements in the following two aspects: This paper will add the indicator of technological innovation. On the one hand, there is no literature on the relationship between the three in China. On the other hand, technological innovation is closely related to financial development and the income gap between urban and rural residents. Thus, the level of technological innovation cannot be abandoned in the study; It will verify the FKC theory and analyze the link between technological innovation and urban-rural income gap under the FKC hypothesis. 

It is found that the long-run relationship exists among the variables under the structural breaks. The main finding obtained from the long-run coefficient estimates reveal that technological innovation increases urban-rural income gap. The findings confirm the validity of the FKC hypothesis for China’s economy in the long run. The causality analysis shows a bi-directional causality between financial development, technological innovation and urban-rural income gap in the long run. We can present the following policy suggestions for China’s economy.

First, expand the scale of rural financial development and improve the efficiency of rural financial development. Because of the difficulty of loans in rural areas and the outflow of rural funds from rural areas through household savings and savings of township enterprises, when farmers have financing needs, they cannot get corresponding financing support because they cannot meet the credit threshold of financial institutions, and the efficiency of rural financial allocation is low. Therefore, the government should establish a sound agricultural financial system, optimize the share of agricultural financial services and financial resources, and try to improve the uneven distribution of financial resources between urban and rural areas and financial development to benefit more high-income people and less rural residents. Increase preferential policy support for rural economic development, improve the rural financial organization system, and promote agricultural modernization. In addition, the continuous expansion and upgrading of rural agricultural financial services will help attract more financial institutions and financial products to enter the countryside, and will also help rural areas reduce financing costs and thresholds, and reduce the loss of rural financial resources and talents. The improvement of rural financial operation efficiency can convert the absorbed rural savings into rural loans in real time, increase support for 'agriculture, rural areas and farmers' funds, increase the utilization rate of rural financial resources, and improve the development of rural financial development.

Second, Financial regulators should appropriately relax the market access threshold, encourage rural commercial banks, small and medium-sized commercial banks, financing guarantee companies, and other institutions to take root in the countryside, form a diversified rural financial service system that is competitive and coexisting with each other, and constantly improve the financial availability of farmers.

Third, adjust the structure of financial development, improve the efficiency of financial development, and narrow the income gap between urban and rural residents. We should constantly improve the financial market, establish a perfect and multi-level financial system that adapts to the development of the times, and set up corresponding regulatory authorities to give full play to the independent effect of the financial market. In terms of the income distribution, we should not only focus on efficiency but also fairness in distribution. So should financial development. We should pay attention to the efficiency and structure of financial development while expanding the scale of financial development. The focus of financial development before was mainly on expanding the scale, and always ignored the efficiency of financial development. Therefore, while developing finance, we should take into account the expansion of development scale and efficiency improvement, but we can not ignore the rationality of financial development structure.

Last, the government should take practical measures and introduce preferential policies to support technological innovation of small and medium-sized enterprises, create good financing conditions for SMEs, and establish and develop technological service systems for SMEs. In addition, local governments should insist on promoting industry to agriculture in technological innovation, cities supporting rural areas, and continuously increasing the intensity of financial investment in science and technology. On the one hand, while creating a good background to promote technological innovation, it should also introduce and implement relevant policies to promote technological innovation, create special innovation funds, actively subsidize and encourage scientific and technological innovation in large, medium, and small enterprises, research and development institutions, universities and other major scientific and technological innovation achievements. Vigorously promote industry-university-research cooperation between enterprises and universities, and help enterprises expand the market for technological innovation products and services. On the other hand, actively establish and improve the agricultural technology innovation coordination mechanism, improve the construction of agricultural technology innovation software and hardware, raise the level of agricultural technology, and lay a solid foundation for the construction of new rural industries.

This paper has some limitations. First, since we strengthen the relationship between technological innovation and urban-rural income gap in the context of FKC hypothesis, several explanatory variables such as globalization, human capital, and renewable energy are not included in our specifications. Second, many indicators of financial developments such as current liabilities and financial development index are not used for empirical analysis. At the same time, the study may inspire future researches. In this paper, there are some limitations in the relationship between patent licensing data analysis and urban-rural income gap. In the follow-up study, independent technological innovation indicators such as high-tech product export and R&D investment can be considered to further demonstrate the impact of technological innovation on urban-rural income gap and the differences in the results of different indicators. Therefore, future researches can investigate the impact of technological innovation on urban-rural income gap in detail and prefer comparative empirical results. 

Thanks once more!

Editors #comments1: 

Response: Thank you very much for your reminder. We have rechecked and corrected the references in the manuscript. The details are as follows:

References：

1. Kuznets S. Economic growth and income inequality. The Economic Review.1995; 45(1),1–28

2. Greenwood J, Jovanovich B. Financial development, growth, and the distribution of income. Journal of Political Economy. 1990; 98(5), 1076–1107.

3. Zhenmei Hou, Maozai Tian, Zhihao Wang, Yan Dou. Study on the impact of technological innovation on the income gap between urban and rural residents—Based on the empirical analysis of Western Ethnic agglomeration. Practice and understanding of mathematics. 2020; 50(02): 53–64.

4. Pan X, Uddin M K, Ai B, Pan X, Saima U. Influential factors of carbon emissions intensity in OECD countries: evidence from symbolic regression. Journal of Cleaner Production. 2019; 220,1194–1201. https://doi.org/10.1016/j.jclepro.2019.02.195.

5. Huiping W, Meixia W. Effects of technological innovation on energy efficiency in China: evidence from dynamic panel of 284 cities. Science of The Total Environment. 2020; 709,136172. https://doi.org/10.1016/j.scitotenv.2019.136172.

6. Yii K-J, Geetha C. The nexus between technology innovation and CO2 emissions in Malaysia: evidence from granger causality test. Energy Procedia. 2017; 105,3118–3124. https://doi.org/10.1016/j.egypro.2017.03.654.

7. Townsend R M, Ueda K. Financial Deepening, Inequality, and Growth: A Model—Based Quantitative Evaluation. The Review of Economic Studies.2006; 73(1):251–293. https://doi.org/10.1111/j.1467-937X.2006.00376.x.

8. Clarke, George RG, Heng-fu Z, Lixin Colin X. Finance and income inequality: test of alternative theories. World Bank Publications. 2003.

9. Pradhan R P. The nexus between finance, growth and poverty in India: The cointegration and causality approach. Asian Social Science. 2010; 6(9):114.

10. Law S H, Tan H B, Azman-Saini W N W. Financial development and income inequality at different levels of institutional quality. Emerging Markets Finance and Trade. 2014; 50(sup1),21–33. https://doi.org/10.1016/j.jclepro.2019.02.195.

11. Sehrawat M, Giri, A K. Financial development and income inequality in India: an application of ARDL approach. International Journal of Social Economics.2015; 42(1):64–81. https://doi.org/10.1108/IJSE-09-2013-0208.

12. Haishu Qiao, Li Chen. Re test of the inverted U-shaped relationship between financial development and urban-rural income gap: An Empirical Analysis Based on China's County cross-sectional data. China's rural economy.2009; (07):68–76+85.

13. Zongyi Hu, Yiwen Liu. Research on Kuznets Effect of unbalanced financial development and urban-rural income Gap Empirical Analysis Based on cross-sectional data of counties in China. Statistical research.2010; 27(05):25–31.doi:10.3969/j.issn.1002-4565.2010.05.004. 

14. Nan Yang, Chuoxin Ma. The dynamic inverted U evolution of the impact of China's financial development on the urban-rural income gap and the prediction of its decline point. Financial research. 2014; (11):175–190.

15. Zhiqiang Ye, Xiding Chen, Shunming Zhang. Can financial development reduce the income gap between urban and rural areas-Evidence from China. Financial research.2011; (02):42–56.

16. Fei Jia. Test on the unbalanced effect of financial development on urban rural income gap. Statistics and decision making. 2015; (24): 92–95.doi:10.13546/j.cnki.tjyjc.2015.24.026.

17. Yongqiang Sun, Yulin Wan. Financial Development, Opening Up and Income Gap between Urban and Rural Residents: An Empirical Analysis Based on Interprovincial Panel Data from 1978 to 2008. Financial Research. 2011; (01):28–39.

18. Youcai Yang. Financial Development and Economic Growth: An Analysis Based on the Threshold Variables of China's Financial Development. Journal of Financial Research. 2014; (02):59–71.

19. Edward E, Leamer. What’s the use of factor contents? Journal of International Economics.2000; 50(1). https://doi.org/10.1016/S0022-19-96(99)00004-5.

20. Antonio C. Agglomeration effects in European. European Economic Review.2002; 46:213-227. https://doi.org/10.1016/S0014-2921(00)00099-4.

21. Cirillo V, Corsi M, D’Ippoliti C. European households’incomes since the crisis. Investigaci6n Economica.2017; 76(301):57–85. https://doi.org/10.1016/j.inveco.2017.12.002

22. Nathan M, Lee N. Cultural Diversity, Innovation, and Entrepreneurship: Firm-level Evidence from London. Economic Geography.2013; 89(4):367–394. 

23. Aghion P, Akcigit U, Bergeaud A, Blundell R, Hémous D. Innovation and top income inequality. The Review of Economic Studies.2019; 86(1),1–45. https://doi.org/10.1093/restud/rdy027.

24. Yong Chen, Zhe Bai. Skill-biased technological progress, labor agglomeration effect and widening of regional wage gap. China Industrial Economy. 2018; (09):79–97. doi:10.19581/j.cnki.ciejournal.2018.09.015.

25. Peng Zeng, Gongliang Wu, Xiaojun Zang. Research on the Relationship between Technological Progress, Urbanization and Urban-Rural Income Gap: Based on empirical analysis of Urban Agglomerations in China. East China Economic Management. 2016; 30(02):64–70.

26. Zhiqing, Dong, Xiao Cai, Linhui Wang. Skill Premium: an explanation based on the direction of technological progress. Chinese Social Sciences. 2014; (10):22–40+205-206.

27. Lei Ma. Human capital structure, technological progress and urban rural income gap: an analysis based on the panel data of 30 provinces in China from 2002 to 2013. East China economic management. 2016; 30(02):56–63.doi:10.3969/j.issn.1007-5097.2016.02.010.

28. Dickey D A, Fuller, W A. Likelihood ratio statistics for autoregressive time series with a unit root. Econometrica.1981; 49(4),1057–1072. https://doi.org/10.2307/1912517.

29. Phillips P C B, Perron P. Testing for a unit root in time series regression. Biometrika.1988; 75(2),335–346. https://doi.org/10.1093/biomet/75.2.335

30. Elliott G, Rothenberg, T J, Stock J H. Efficient tests for an autoregressive unit root. Econo-metrica.1996; 64(4),813–836. https://doi.org/10.2307/2171846.

31. Pesaran M H, Shin Y, Smith R. Bounds testing approaches to the analysis of level relationship. Journal of Applied Econometrics, 2001; (3):289-326. https://doi.org/10.1002/jae.616.

32. Stock J H, Watson M W. A simple estimator of cointegrating vectors in higher order integrated systems. Econometrica.1993; 61(4),783–820. https://doi.org/10.2307/2951763.

33. Phillips P C B, Hansen B E. Statistical inference in instrumental variables regression with I(1) processes. The Review of Economic Studies. 1990; 57(1),99–125. https://doi.org/10.2307/2297545.

34. Park J Y. Canonical cointegrating regressions. Econometrica.1992; 60(1),119–143. http://www.jstor.org/stable/2951679.

35. Engle R F, Granger C W J. Co-integration and error correction: representation, estimation, and testing. Econometrica.1987; 55(2), 251–276. https://doi.org/10.2307/1913236.

36. Jing Wang. An analysis of the relationship between Taiwan's economic openness and income gap—An Empirical Study Based on ardl-ecm model. Industrial economic circles. 2016; 03(02):164–172.

37. Bildirici M E. The effects of militarization on biofuel consumption and CO2 emission. Journal of Cleaner Production.2017; 152,420–428. https://doi.org/10.1016/j.jclepro.2017.03.103.

38. Fan Ye, Dongtao Zou, Xiheng Yuan. The impact of economic financialization on the income gap between urban and rural areas in China: an analysis based on provincial panel data from 1978 to 2013. Contemporary economic science. 2015; 37 (06):61–67+124.

39. Aghion P, Akcigit U, Bergeaud A, Blundell R, Hemous D. Innovation and top income inequality. Review of Economic Studies.2019; 86(1),1–45.https://doi.org/10.1093/restud/rdy027.

40. Mnif S. Bilateral relationship between technological changes and income inequality in developing countries. Atlantic Review of Economics.2016; 1(1),4.

41. Yingli Zhang, Zhengyong Yang. Mechanism and dynamic analysis of financial development and urbanization on urban and rural income gap. Statistics and Decision Making. 2018; 34(05):84-88. doi:10.13546/j.cnki.tjyjc.2018.05.020.

42. Antonelli C, Gehringer A. Technological change, rent and income inequalities: a Schumpeterian approach. Technological Forecasting and Social Change.2017; 115, 85–98. https://doi.org/10.1016/j.techfore.2016.09.023.

43. Zhihuan Chen, Hongge Zhu, Bo Cao. Financial Development, Economic Growth, and Energy Consumption Analysis is based on the ARDL-ECM model. Technical Economy and Management Research. 2020; (05):97–104.

44. Tao Feng, Maoguang Wu, Meisha Zhang. Financial development, industrial structure and urban-rural income gap: an analysis based on the perspective of "from real to virtual" finance. Exploration of economic problems. 2020; (10):170–181.

45. Shulan Fei, Jiqiang Guo. The impact of the statistical income attribution of migrant workers on the income gap between urban and rural areas. Statistical study. 2014; 31(06):17–24.

46. Jinpei Liu, Xiaoxia Song, Huayou Chen, Guanzhen Wang, Zhen Wang. Study on long-term equilibrium and causal dynamics of influencing factors affecting per capita carbon emissions in China by based on the empirical analysis of structural mutation ARDL-VECM model. Operations Preparation and Management. 2019; 28(09):57–65.

47. Ping Li, Tinghua Liu. Research on the Relations between Technology Innovation and Income Inquality—Based on China Data. Industrial Technical economy. 2009; 1:41–46.

48. Jingwen Li, Zhengdong Yang. The impact of technological innovation on industrial structure and income distribution. Journal of Social Science of Jilin University. 2013; (6):5–11.doi:10.15939/j.jujsse.2013.06.014.

49. Madhu S H W, Giri A K. Financial development and income inequality in India: an application of ARDL approach. Social Indicators Research.2011; 41(2):1-26. https://doi.org/10.1108/IJSE-09-2013-0208.

50. Huaizhong Mu, Peng Wu. Urbanization, industrial structure optimization and urban and rural income gap. Economist. 2016; (05):37–44.doi: 10.16158/j.cnki.51-1312/f.2016.05.006.

51. Pece A M, Simona O E O, Salisteanu F. Innovation and economic growth: an empirical analysis for CEE countries. Procedia Economics and Finance.2015; 26,461–467. https://doi.org/10.1016/S2212-5671(15)008746.

52. Permana M Y, Lantu D C, Suharto, Y. The effect of innovation and technological special-ization on income inequality. Problems and Perspectives in Management.2018; 16(4):51–63. 

53. Risso W A, Carrera E J S. On the impact of innovation and inequality in economic growth. Economics of Innovation and New Technology. 2019; 28(1):64–81. https://doi.org/10.1080/10438599.2018.1429534.

54. Sohag K, Begum R A. Abdullah, S.M.S. et al. Dynamics of energy use, technological innovation, economic growth and trade openness in Malaysia. Energy.2015; 90 (Part2):1497–1507. https://doi.org/10.1016/j.energy.2015.06.101.

55. Shahbaz M, Raghutla C, Song M, Zameer, H, Jiao Z. Public-private partnerships investment in energy as new determinant of CO2 emissions: the role of technological innovations in China. Energy Economics.2020; 86,104664. https://doi.org/10.1016/j.104664.

56. Thornton J, Tommaso C D. The long-run relationship between finance and income inequal-ity: evidence from panel data. Finance Research Letters. 2020; 32,101180. https://doi.org/10.1016/j.frl.2019.04.036.

57. Yantao Zhang. Energy Consumption and Economic Growth Restudy—based on FMOLS, DOLS, CCR Research. Journal of North China University of Electric Power (Social Sciences). 2012; (01):5–9.

58. Cetin M, Demir H, Saygin S. Financial development, technological innovation and income inequality: Time series evidence from Turkey. Social Indicators Research.2021; 156(1),47–69. https://doi.org/10.1007/s11205-021-02641-7.

59. Ozturk S, Cetin M, Demir H. Income inequality and CO2 emissions: nonlinear evidence from Turkey. En-vironment, Development and Sustainability.2021; 1–18. https://doi.org/10.1007/s10668-021-01922-y.

60. Altunbaş Y, Thornton J. The impact of financial development on income inequality: A quantile regression approach. Economics Letters.2019; 175,51–56. https://doi.org/10.1016/j.econlet.2018.12.030.

Thanks!

---

## [Decision Letter · Decision Letter 2]

5 Dec 2022

Financial development, technological innovation and urban-rural income gap: time series evidence from China

PONE-D-22-09786R2

Dear Dr. Wu,

We’re pleased to inform you that your manuscript has been judged scientifically suitable for publication and will be formally accepted for publication once it meets all outstanding technical requirements.

Kind regards,

Ming Zhang, Ph.D.

Academic Editor

PLOS ONE

Additional Editor Comments (optional):

Reviewers' comments:

Reviewer's Responses to Questions

**Comments to the Author**

1. If the authors have adequately addressed your comments raised in a previous round of review and you feel that this manuscript is now acceptable for publication, you may indicate that here to bypass the “Comments to the Author” section, enter your conflict of interest statement in the “Confidential to Editor” section, and submit your "Accept" recommendation.

Reviewer #1: All comments have been addressed

Reviewer #2: All comments have been addressed

2. Is the manuscript technically sound, and do the data support the conclusions?

Reviewer #1: Yes

Reviewer #2: Yes

3. Has the statistical analysis been performed appropriately and rigorously? 

Reviewer #1: Yes

Reviewer #2: Yes

4. Have the authors made all data underlying the findings in their manuscript fully available?

Reviewer #1: Yes

Reviewer #2: Yes

5. Is the manuscript presented in an intelligible fashion and written in standard English?

Reviewer #1: Yes

Reviewer #2: Yes

6. Review Comments to the Author

Reviewer #1: The review comments are well-addressed. The present version of the manuscript looks much better in its present form.

Well Done !!!

Reviewer #2: All requests have been fulfilled by the authors. The article is well designed. Appropriate answers are given to the referee's suggestions. Although the methodology used is not very up-to-date, it is used well in harmony with the theory.

7. PLOS authors have the option to publish the peer review history of their article (what does this mean?). If published, this will include your full peer review and any attached files.

Reviewer #1: No

Reviewer #2: No

---

## [Editor Report · Acceptance letter]

1 Feb 2023

PONE-D-22-09786R2 

Financial development, technological innovation and urban-rural income gap: time series evidence from China 

Dear Dr. Wu:

I'm pleased to inform you that your manuscript has been deemed suitable for publication in PLOS ONE. Congratulations! Your manuscript is now with our production department. 

Kind regards, 

on behalf of

Dr. Ming Zhang 

Academic Editor

PLOS ONE